# Partially native intermediates mediate misfolding of SOD1 in single-molecule folding trajectories

Supratik Sen Mojumdar[1], Zackary N. Scholl[1], Derek R. Dee[1], Logan Rouleau[1], Uttam Anand[1], Craig Garen[1] & Michael T. Woodside [1,2]

Prion-like misfolding of superoxide dismutase 1 (SOD1) is associated with the disease ALS, but the mechanism of misfolding remains unclear, partly because misfolding is difficult to observe directly. Here we study the most misfolding-prone form of SOD1, reduced un-metallated monomers, using optical tweezers to measure unfolding and refolding of single molecules. We find that the folding is more complex than suspected, resolving numerous previously undetected intermediate states consistent with the formation of individual β-strands in the native structure. We identify a stable core of the protein that unfolds last and refolds first, and directly observe several distinct misfolded states that branch off from the native folding pathways at specific points after the formation of the stable core. Partially folded intermediates thus play a crucial role mediating between native and non-native folding. These results suggest an explanation for SOD1's propensity for prion-like misfolding and point to possible targets for therapeutic intervention.

[1] Department of Physics, University of Alberta, Edmonton, AB, Canada T6G 2E1. [2] National Institute for Nanotechnology, National Research Council, Edmonton, AB, Canada T6G 2M9. Correspondence and requests for materials should be addressed to M.T.W. (email: michael.woodside@ualberta.ca)

Cu/Zn-superoxide dismutase 1 (SOD1) is a ubiquitous cytosolic antioxidant that protects cells against damage from superoxide[1]. How SOD1 folds is of particular interest because its misfolding has been linked to amyotrophic lateral sclerosis (ALS), a fatal neurodegenerative disorder affecting motor neurons[1–4]. Over 100 mutations in SOD1 are associated with the familial form of ALS[3, 4], and aggregates of misfolded SOD1 within motor neurons are suspected as pathogenic agents in ALS[3–6], similar to the role of aggregates in the context of other protein misfolding diseases[7, 8]. Moreover, misfolded SOD1 can convert natively folded protein and propagate misfolding from cell to cell[9–11], suggesting that it is capable of prion-like propagation. However, the mechanisms by which misfolding and conversion take place—and their relation to native folding—remain unclear, in part because it is difficult to observe rare and transient misfolding events directly.

The structure and folding of SOD1 have been studied extensively with ensemble biophysical techniques to elucidate the properties of native and non-native states. Natively, SOD1 adopts an immunoglobulin-like, β-barrel fold (Fig. 1a), forming a homodimer in which each subunit consists of eight antiparallel β-strands and two major loops, binding one $Cu^{2+}$ and one $Zn^{2+}$ ion in the active site region[12, 13]. Catalysis occurs at the Cu-binding site, whereas Zn binding enhances the structural stability[12]. Dimeric holo-SOD1 is very stable and soluble in solution, but removing the metal ions greatly reduces its stability[14–17]. Reduction of the native disulfide bond in SOD1[12, 13] not only destabilizes the dimer[18] but also enhances the propensity to misfold, with reduced apo-SOD1 forming fibrillar aggregates

in vitro similar to other amyloidogenic proteins[14, 19]. Native structure is generally believed to form via a three-state process, with each monomer subunit folding in a two-state process before dimerization[20–24], although some studies have suggested a more complex process involving intermediates[25–27]. Because of the high stability and aggregation resistance of dimeric holo-SOD1, de-metallation and dimer dissociation have been proposed to be key events in SOD1 misfolding and aggregation[3–5, 15, 27–29], and monomeric apo-SOD1 has been used as a model in numerous biophysical studies of SOD1 misfolding and aggregation in vitro[24, 27, 30–32]. Indeed, apo-SOD1[2SH] has even been proposed to be toxic in vivo[33].

The structural properties of misfolded forms of SOD1 have begun to be studied[34, 35], but the nature of the misfolding mechanism, the role of intermediate states (if any), and the identity of the toxic species all remain controversial[24, 27, 30]. Even though native wild-type (wt) SOD1 is quite resistant to misfolding and aggregation[15], it can aggregate both in vivo[36]. and in vitro[37], indicating an inherent ability to misfold even without mutation. Exposure of the hydrophobic β-barrel core, possibly through partially structured intermediates, has been proposed to drive misfolding and aggregation[27, 30]. However, although intermediates playing such a role have been suggested by simulations[26] and NMR measurements[27], other experiments found no such evidence for intermediates[20–24], raising questions about their existence and role.

Single-molecule approaches are well-suited to address the challenges posed by misfolding, because they allow the trajectory of a molecule between different conformations to be followed in

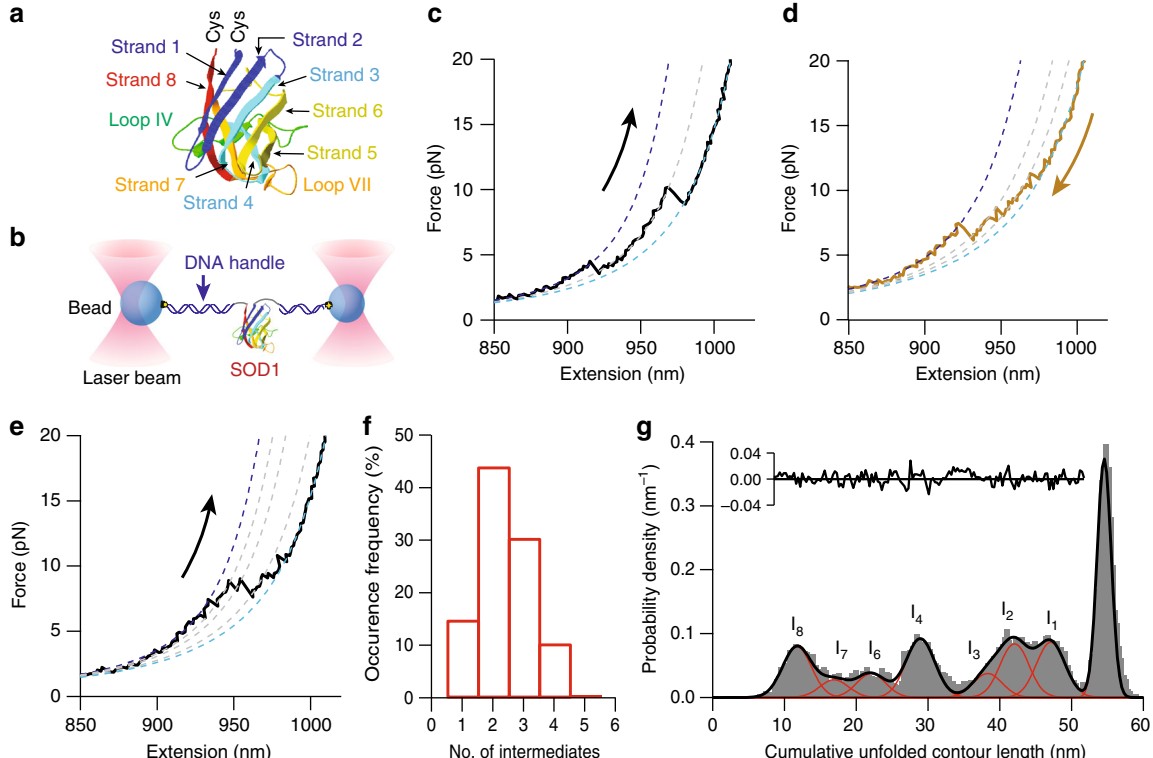

**Fig. 1** Force spectroscopy of SOD1. **a** The structure of the apo-SOD1 monomer, showing the 8 β-strands and two functional loops (PDB ID: 3ECU). Cys residues engineered at the termini were used for attaching DNA handles. **b** Schematic of the force spectroscopy assay. The SOD1 monomer was attached covalently to DNA handles linked to polystyrene beads held by laser beams. **c** An unfolding FEC of the SOD1 monomer showing an intermediate state. Dashed lines: WLC fits to the folded (blue), intermediate (gray), and unfolded (cyan) states. **d** A refolding FEC showing two intermediate states. **e** An unfolding FEC with three intermediate states. **f** The distribution of the number of intermediate states observed in unfolding FECs. **g** The distribution of the cumulative contour length of unfolded protein at each state observed in the unfolding FECs, as measured from the natively folded state via WLC fits. The positions of seven intermediate states were identified by Gaussian fits (red) of the peaks in the distribution

**Table 1 States observed during native unfolding and refolding of monomeric apo-SOD1**

| State | Unfolding $\Delta L_c$ (nm) | Refolding $\Delta L_c$ (nm) | Proposed structure | Expected $\Delta L_c$ (nm) |
|---|---|---|---|---|
| U | $54.7 \pm 0.3$ | $54.8 \pm 0.3$ | Unfolded | 54.2 |
| $I_1$ | $47 \pm 1$ | $47 \pm 1$ | Strands 2–3 and loop II, or Strands 1–2 and loop I | 46.7 or 47.4 |
| $I_2$ | $42 \pm 1$ | $42 \pm 1$ | Strands 2–4 and loops II–III, or Strands 1–3 and loops I–II | 43.7 or 44.2 |
| $I_3$ | $38 \pm 1$ | $38 \pm 1$ | Strands 1–4 and loops I–III | 39 |
| $I_4$ | $29 \pm 1$ | $30 \pm 1$ | Strands 2–5 and loops II–IV | 28.5 |
| $I_5$ | — | $26 \pm 1$ | Strands 2–6 and loops II–IV, or Strands 1–5 and loops I–IV | 25.3 or 25.3 |
| $I_6$ | $22 \pm 1$ | $21 \pm 1$ | Strands 1–6 and loops I–V | 19.4 |
| $I_7$ | $17 \pm 1$ | — | Strands 2–7 and loops II–VI | 17 |
| $I_8$ | $12 \pm 1$ | $11 \pm 2$ | Strands 1–7 and loops I–VI | 14 |
| F | 0 | 0 | Strands 1–8 and loops I–VII | 0 |

All contour lengths represent the cumulative length change measured from the natively folded state. All values expressed as mean ± s.e.m.

real time[38]. As a consequence, structural sub-populations—including intermediates in a heterogeneous ensemble—can be distinguished and characterized directly, and the rare and/or transient states that are commonly associated with misfolding can be identified and probed[39–42]. Single-molecule methods have been used to study prion proteins[39, 43–46] and other proteins with prion-like properties such as α-synuclein[40, 47–50] and Aβ[50, 51], as well as to study disulfide-bond isomerization in SOD1[52], but they have not yet been applied to study misfolding in SOD1.

Here we examine SOD1 folding using single-molecule force spectroscopy (SMFS), applying force to the ends of individual SOD1 molecules with optical tweezers to destabilize the structure and then monitoring changes in the molecular extension as the conformation changes in response to the load[53]. We focus on monomeric, reduced apo-SOD1 (apo-SOD1$^{2SH}$) as the form of the protein that is least stable and most prone to misfolding[20, 21, 32, 54, 55], aiming to map the pathways for both native folding and misfolding, and determine how they are related. We find from these high-resolution SMFS measurements that native folding is more complex than previously thought, involving multiple intermediate states that are consistent with the sequential formation of each strand in the native fold. Identifying several distinct misfolded states, we elucidate a misfolding mechanism whereby misfolded states branch from the native pathway at specific intermediates, after the formation of a stable, native-like core.

## Results

**Force–extension curves show primarily native folding**. Monomeric human SOD1, mutated to remove solvent-exposed cysteines[20, 21] and disrupt the dimer interface[20, 21, 56], was attached to DNA handles bound to polystyrene beads held in dual-trap tweezers (Fig. 1b), and the extension of the protein–DNA construct was measured as the traps were moved apart and back together at constant speed to ramp the force up and down. The force was ramped up until the protein was completely unfolded (Fig. 1c, black), and then back down to near zero for 0.5–10.5 s to allow the protein to refold. Unfolding force–extension curves (FECs) showed a non-linear rise of force with extension—as is typical for polymers stretched under tension—interrupted by one or more "rips" (where the extension jumped abruptly concomitant with a drop in the force) characteristic of unfolding transitions, resulting in sawtooth patterns in the FECs (Fig. 1c, black). The protein was completely unfolded by only ~10 pN, a low force consistent with the low stability of the apo state.

To determine whether the protein was natively folded in each FEC, the total contour length change ($\Delta L_c$) during the unfolding

transitions was measured by fitting the initial segment of the FEC before any rips (the folded state) and the final segment after all the rips (the unfolded state) to extensible worm-like chain (WLC) models[57] for the DNA handles and unfolded polypeptide (Fig. 1c, dotted lines), as described previously[39]. Analyzing 3050 FECs from 12 molecules, the vast majority (>95%) showed a total $\Delta L_c$ upon unfolding of $54.7 \pm 0.3$ nm (all errors represent s.e.m. unless otherwise specified), consistent with the value expected from the native structure of apo-SOD1 as found by crystallography[58], $\Delta L_c$ = 54.2 nm. Refolding FECs (Fig. 1d, red) demonstrated qualitatively similar features, with most showing the same value for the total $\Delta L_c$ upon refolding, $\Delta L_c = 54.8 \pm 0.3$ nm. In most of the FECs, the protein was thus unfolding from, and refolding into, the native structure.

**Folding occurs through multiple intermediate states**. A notable feature of these curves is that they all invariably contain at least two rips, indicating the presence of obligate metastable intermediate states. In a few cases, a single intermediate was observed (as in Fig. 1c), but most often more were seen (Fig. 1e). The number of intermediates varied between 1 and 5 from one curve to the next, with 2 or 3 being most common (Fig. 1f). FECs often displayed instances of hopping back and forth between intermediates (as in Fig. 1d, e), indicating that the unfolding and refolding transitions were near equilibrium, a fact also reflected in the similarity between the forces for unfolding and refolding. Intriguingly, the FECs were quite heterogeneous: the same contour lengths were not always seen from one curve to the next, but certain lengths recurred over repeated pulls, suggesting that each FEC sampled only a subset of the full set of intermediates that could be formed by apo-SOD1 monomers. The folding dynamics were thus considerably more complex than the simple two-state model of monomer folding proposed in previous ensemble studies[20–24].

To characterize the intermediates more fully, we used WLC fits of the intermediates in the FECs to determine $\Delta L_c$ for each transition, and we plotted the distribution of the cumulative $\Delta L_c$ as measured from the folded state (reflecting the length of polypeptide unfolded) for every state observed in all FECs (Fig. 1g). Examining the cumulative $\Delta L_c$ from the folded state allowed the whole set of possible intermediates to be discerned, even though not every intermediate was observed in each FEC: values that recurred in different pulls were evident as peaks in the distribution, with each peak corresponding to a distinct intermediate state. In addition to the peak representing complete unfolding of the monomer, at 54.7 nm (present in all curves), four peaks of roughly equal width (~ 4 nm) can be clearly seen at ~12 nm, 29 nm, 42 nm, and 47 nm (Fig. 1g, red), identifying the

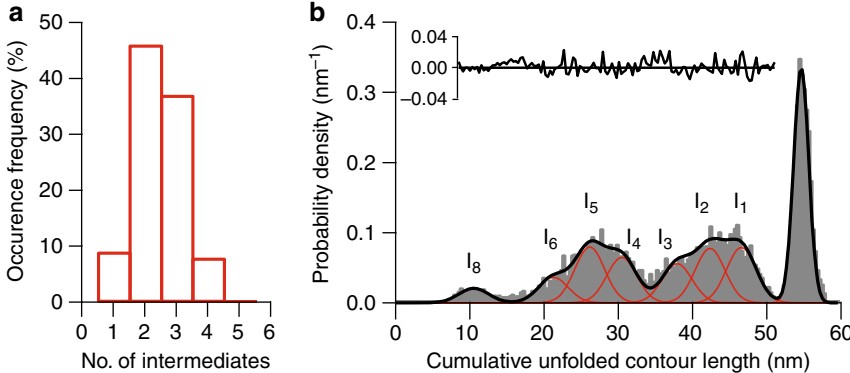

**Fig. 2** Refolding intermediates. **a** The distribution of the number of intermediate states observed in refolding FECs. **b** The distribution of the cumulative contour length of unfolded protein at each state observed in the refolding FECs, as measured from the natively folded state via WLC fits. The positions of seven intermediate states were identified by Gaussian fits (red) of the peaks in the distribution, all but one matching intermediates seen during unfolding

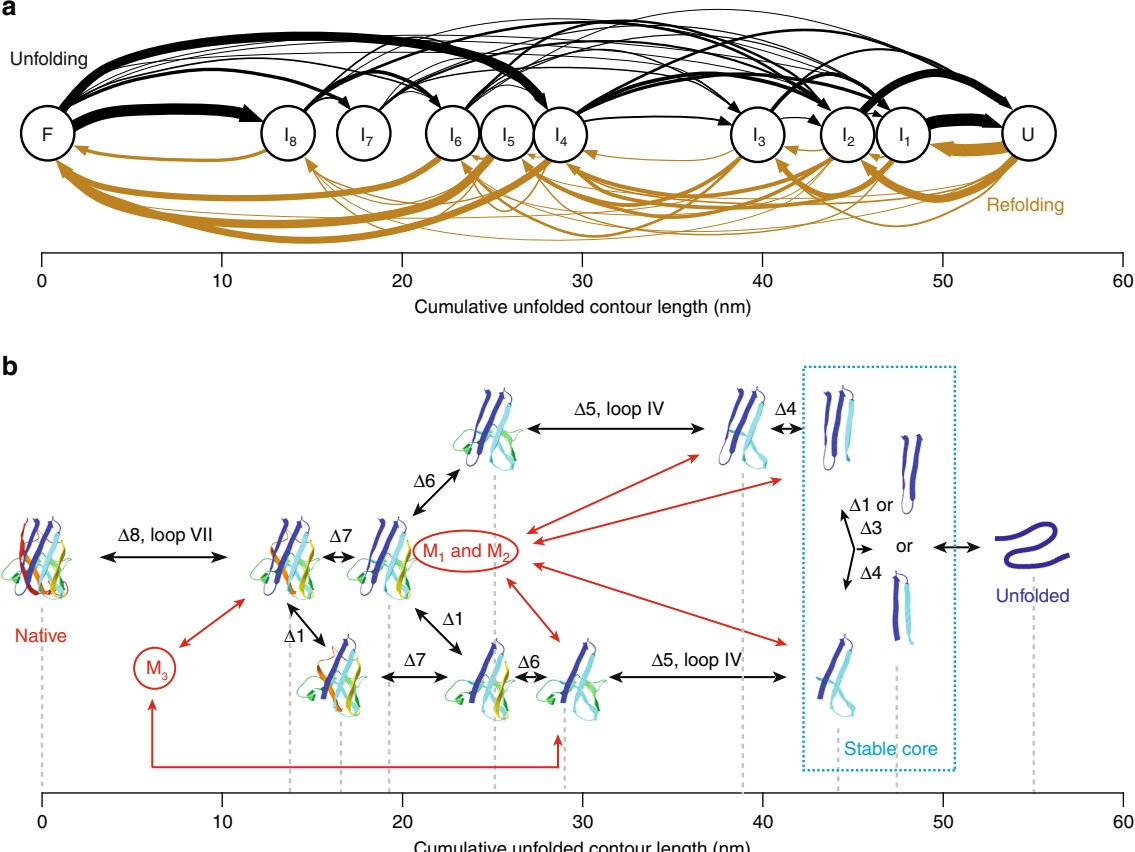

**Fig. 3** Folding pathways of monomeric SOD1. **a** Map of the experimentally observed transitions between intermediates in FECs showing native unfolding (black) and refolding (brown). Intermediates assigned based on the cumulative $\Delta L_c$ values. Arrows reflect all pairwise transitions between states, with line thickness representing the relative frequency of the transition. **b** Proposed structures of the intermediate states for native unfolding/refolding and dominant pathways between them (black), as deduced from a network analysis of the transitions. The intermediates have cumulative unfolded contour lengths consistent with the sequential removal/addition of individual β-strands in the native structure. The stable core that unfolds last and refolds first is indicated by the box. Misfolded states branch off from the native pathways (red) after formation of the stable core, indicating that partially native intermediates mediate the misfolding

length of the four most common intermediates. Assuming that the peaks for all intermediates have similar widths, we fit the remainder of the distribution to three additional Gaussians, peaked in turn at 17 nm, 22 nm, and 38 nm (Fig. 1g, red). The residual after fitting the distribution to these Gaussians (Fig. 1g,

inset) was close to zero. The positions of all peaks are listed in Table 1.

A similar analysis for refolding FECs again showed a variable number of intermediate states in each curve (Fig. 2a). The distribution of cumulative $\Delta L_c$ values measured from the folded

state (Fig. 2b, black) was fit to Gaussians of the same width as for the unfolding FECs (Fig. 2b, red) to identify the set of all intermediates observed during refolding. The fit residual was again close to zero (Fig. 2b, inset). The fitting revealed peaks at many of the same lengths as for unfolding (Table 1). Indeed, in all cases the peaks were the same within error for both unfolding and refolding, with two exceptions: a peak at 26 nm was observed during refolding that was not seen during unfolding, and the 17-nm intermediate seen in unfolding FECs was not well-defined during refolding. The concordance between the intermediates seen during unfolding and refolding indicates that effectively the same set of intermediate states was being probed in both unfolding and refolding FECs.

**Mapping the native folding pathways**. To gain a better understanding of the pathways followed between the native fold (N) and the unfolded state (U), we assigned each state in the unfolding and refolding FECs to N, U, or one of the eight different intermediates identified in Figs. 1g and 2b ($I_1$ through $I_8$, in order of increasing amount of folded protein), based on the value of the cumulative $\Delta L_c$ in that state. We then logged all of the pairwise transitions between states observed in the FECs to create transition maps for unfolding (Fig. 3a, black) and refolding (Fig. 3a, brown). These maps show the frequency with which particular paths through the intermediates were observed. Although substantial variability is seen for both unfolding and refolding, the patterns of transitions during unfolding and refolding are qualitatively similar. One of the most noticeable differences between unfolding and refolding is that the final stages of refolding appear to be more cooperative than the initial stages of unfolding: the last intermediate before complete refolding is most commonly $I_5$, whereas the first intermediate during unfolding is most commonly $I_8$.

We next sought to identify possible structures for the intermediates, following the approach used in previous SMFS studies[41, 59] of comparing the length changes observed experimentally to those expected from unfolding different portions of the native structure. Assuming that all loops in SOD1 are unstable and that strands are stable only if fully folded, we considered all possible combinations of folded and unfolded strands consistent with the topology of the native fold (Supplementary Table 1), including not only cases where strands were peeled from the edges of the structure but also those where the native fold ruptured internally (leading to states with disjunct structural elements). For each experimental FEC, we compared the observed cumulative $\Delta L_c$ values to those expected for every structure in Supplementary Table 1, to catalog all structures consistent with each observed intermediate, and then we identified the sequence (s) of structures (Supplementary Fig. 1) that could plausibly account for the observed sequence of transitions in that FEC (see "Methods"). Repeating this analysis across all FECs, we found a minimal set of structures for $I_1$–$I_8$ that was consistent with the great majority (~86%) of FECs, involving the sequential unfolding of single strands (Fig. 3b). The remaining FECs inconsistent with this model might reflect additional pathways, or possibly incorrect state assignments arising from experimental uncertainty in the FEC fits. However, no single extra pathway could add more than 1% to the success rate for the model (Supplementary Fig. 2). We therefore concluded that the model in Fig. 3b represents the most likely intermediates in native folding.

To test this picture, we investigated mutants of SOD1 truncated to match some of the proposed intermediate structures. Because truncation mutants of SOD1 can be highly prone to misfolding and aggregation[60], we focused on the intermediates $I_1$ and $I_2$, which according to our model should be able to fold stably

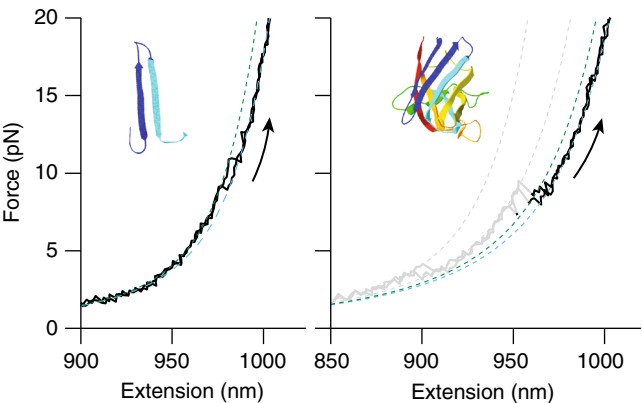

**Fig. 4** FECs of β2–3 hairpin peptide. A peptide containing the residues in strands 2 and 3 folds into a stable structure whose unfolding (left) is similar to that of intermediate $I_1$ in SOD1 (right), consistent with the proposed structure for $I_1$ in Fig. 3b. Dashed lines: WLC fits to the $I_1$ intermediate (green), other intermediate (gray), and unfolded (cyan) states. Average contour length change for unfolding the hairpin peptide: $8.3 \pm 0.8$ nm (246 FECs from four molecules), compared to $7.7 \pm 0.3$ nm for unfolding $I_1$ (1346 FECs from 11 molecules)

and independently. A peptide consisting of strands 1–3 (state $I_2$) proved too aggregation prone for measurement. However, one containing the hairpin made from strands 2 and 3 was soluble, formed a β-structure consistent with the expected hairpin fold (as confirmed by CD spectroscopy, Supplementary Fig. 3), and yielded FECs with unfolding transitions very similar to those for state $I_1$ (Fig. 4), showing that $I_1$ is consistent with the hairpin β2–3. We also compared the experiments on SOD1 to computational simulations of the pulling measurements, using an approach demonstrated recently on mutant SOD1[61]. Coarse-grained modeling of the unfolding under a constant loading rate (see Supplementary Methods) found that unfolding occurred via multiple intermediates, as in the experiments (Fig. 5a). The structures of the intermediates identified in the simulation matched those obtained from our analysis of the FECs (Fig. 3b), and the transition map of the pathways followed during the unfolding simulations (Fig. 5b) agreed very well with the experimental transition map (Fig. 3a), further validating the minimal model of the folding in Fig. 3b.

**Misfolded states of the monomer**. The behavior described above, encompassing the vast majority of all curves, reflected the properties of native folding: SOD1 typically formed its native fold fairly rapidly, while still under tension, after passing through multiple short-lived intermediate states. However, some refolding FECs ended at total $\Delta L_c$ values that were significantly smaller than the 54.7 nm observed for natively folded SOD1 (Fig. 6a, brown). Shorter-than-native length changes were also observed in some of the unfolding FECs (Fig. 6a, black), indicating that this state persisted during the delay time between successive FECs. The protein was thus folding into, and being unfolded from, a structure that was different from the native fold. Varying the delay time between FECs from 0.5 s to 10.5 s, we found that the fraction of unfolding curves that displayed non-native length changes fell from 5 to 1% (Fig. 6b). A single-exponential fit revealed a lifetime of ~6 s for the non-native states, roughly 100-fold longer than the lifetimes of the on-pathway intermediates seen during native unfolding/refolding and over tenfold greater than the time normally required to reach the native state in refolding FECs.

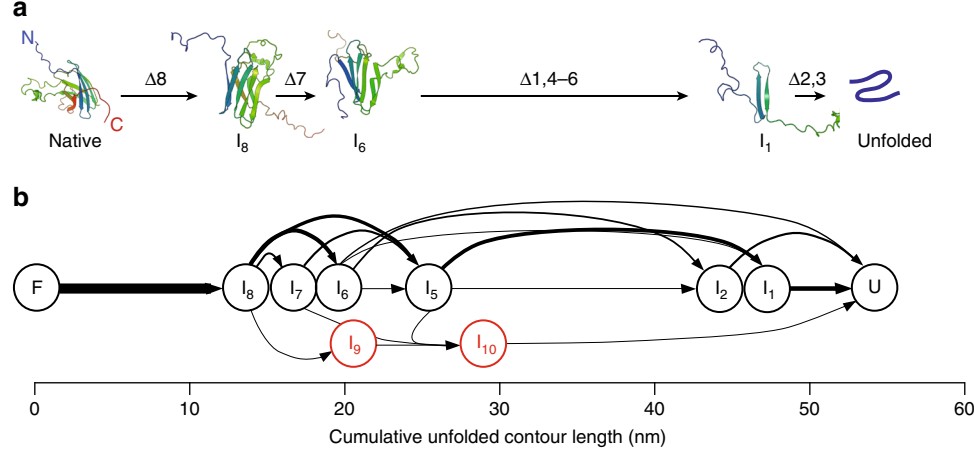

**Fig. 5** Simulations of mechanical unfolding of SOD1. **a** Structural snapshots from a steered molecular dynamics simulation depicting intermediates during SOD1 unfolding. **b** Transition map from the simulations ($N = 26$) showing all pairwise transitions between states; arrow width represents the relative transition probability. The intermediates observed computationally match the structures proposed for those observed experimentally, except for states $I_9$ (β3–7 folded) and $I_{10}$ (β3–6 folded), which were not consistent with the experiments

The properties observed in this minority of FECs—states with non-native contour lengths and long lifetimes that inhibit native folding—reflect the hallmarks of misfolding. Misfolding generally involves intermediates that are non-productive (off the pathway leading to the native fold) and act as kinetic traps; being separated from the native state by large barriers, misfolded states greatly delay achievement of the native fold[62]. We therefore identified the persistent states with non-native contour lengths as misfolded. Looking at the distribution of cumulative $\Delta L_c$ values measured from the folded state for all curves showing non-native behavior (Fig. 6c, Supplementary Fig. 4), we found three peaks, pointing to the existence of at least three different misfolded states with distinct structural properties, which we denoted $M_1$, $M_2$, and $M_3$ (Table 2).

The FECs of misfolded SOD1 still demonstrated some of the same features seen in native folding, however, such as the presence of one or more intermediates in all curves. To compare to the results seen for native unfolding and refolding, we examined the distribution of the cumulative $\Delta L_c$ measured with respect to the location where the native state would normally be found (revealing the length of unfolded protein) for all intermediates in FECs containing misfolding (Fig. 6d). Interestingly, we found peaks at positions similar to those in Figs. 1g and 2b ($15 \pm 2$ nm, $30 \pm 1$ nm, $38 \pm 1$ nm, $42 \pm 1$ nm, $47 \pm 1$ nm). The fact that the high-force intermediates have the same lengths during both native folding and misfolding suggests that misfolding occurs by branching off from the native pathway at one of the on-pathway intermediates. To determine where these branching points occur, we analyzed the distribution of contour length changes for the last intermediate state that occurred before formation of a misfolded state during refolding FECs, and for the first intermediate seen when coming out of the misfolded state during unfolding FECs (Fig. 6e). Peaks were seen at $15 \pm 2$ nm, $30 \pm 1$ nm, $39 \pm 1$ nm, and $44 \pm 1$ nm, which indeed matched the length of specific intermediates on the native folding pathway, identifying these intermediates as the likely branching points for SOD1 misfolding (Table 2).

## Discussion

The picture of the most likely sequence of steps followed during SOD1 folding assembled above is that the first part of the protein to unfold—and the last to refold—is strand 8 and loop VII (the "electrostatic loop"), which form an important part of the dimer interface. This region of the protein has previously been identified both experimentally[27, 30] and computationally[18] as one of the least stable parts of the protein, supporting our interpretation of the identity of the lowest-force intermediate. Next to unfold are strands 7 and/or 1, followed by strand 6, strand 5 and loop IV (the metal-binding loop), and then strand 4. Last to unfold, and first to refold, is a simple β hairpin consisting of strands 2 and 1 or 3. This analysis suggests that there is a distinct hierarchy in the folding of SOD1: a "stable core" consisting of strands 1–3 or 2–4 forms first, then the rest of the structure forms around this core, with the electrostatic loop and dimer interface forming last.

Intriguingly, the overall architecture of the folding is similar to what has been deduced previously from ensemble measurements of apo-SOD1 folding via phi-value analysis[30], despite the fact that we observed multiple intermediates in contrast to the two-state behavior seen in most ensemble measurements[20–24]. The phi values suggested that the transition state includes native-like structure in strands 1–4, remarkably similar to the stable core identified above[30]. Other studies of apo-SOD1 monomers also suggested that the three N-terminal strands are the most stable part of the protein and thus likely to be unfolded last: molecular dynamics simulations predicted that strands 1–3 and strand 7 were the most stable, outlasting the remaining strands and leading to a partially unfolded structure[18], whereas nuclear spin relaxation measurements identified an excited state containing native-like strands 1–3 and 6 amid substantial perturbations in the rest of the structure[27]. The folding mechanism observed in our single-molecule measurements thus parallels what was deduced from previous ensemble studies and simulations, but the higher resolution measurements reveal greater detail via direct observation.

Comparing the single-molecule and ensemble results underscores the prominent difference in folding cooperativity observed. This difference may arise in part because single-molecule assays are generally much more sensitive to transient intermediate states[40, 41]. The different modes of action of force and chemical denaturants may also play a role, with the localized nature of force as a denaturant acting to make the structural changes less global and thus less cooperative. However, the application of force does not habitually induce non-cooperative behavior—two-state folding is commonly seen in SMFS[43, 63, 64]—nor does the use of

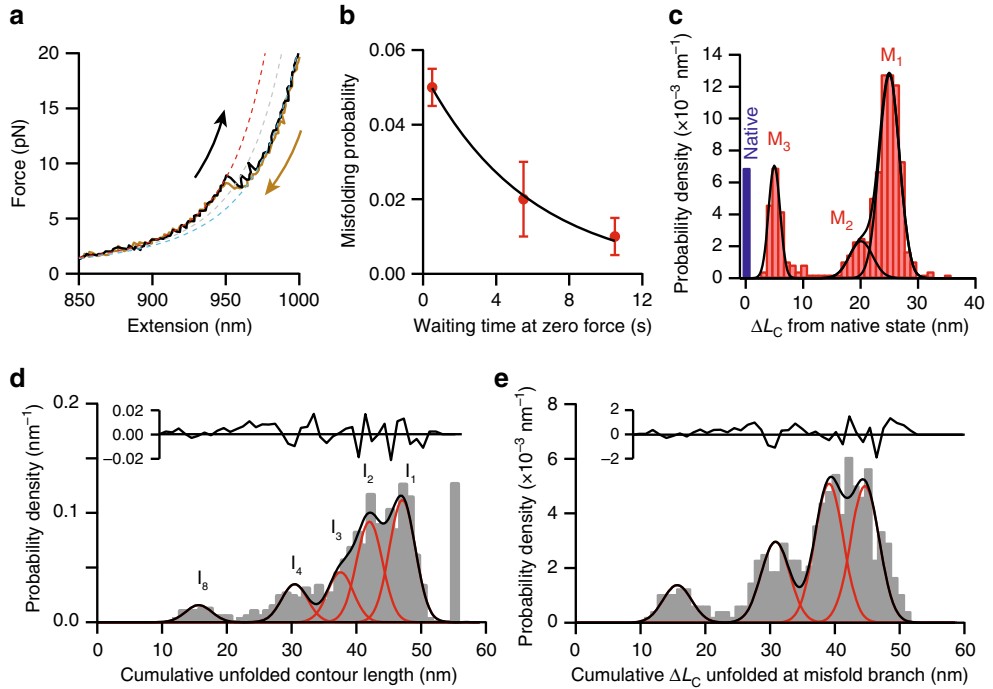

**Fig. 6** SOD1 misfolding. **a** Unfolding (black) and refolding (brown) FECs of a SOD1 monomer showing a non-native total contour length change. Dashed lines: WLC fits to the misfolded (red), intermediate (gray), and unfolded (cyan) states. **b** The fraction of curves showing misfolding decreases exponentially with the waiting time at zero force between successive pulls, revealing a misfolded-state lifetime of 6 s. **c** The distribution of the contour length remaining unfolded in the misfolded state, as measured from all FECs exhibiting misfolding, contains three peaks reflecting at least three distinct misfolded structures. **d** The distribution of cumulative contour length of unfolded protein at each intermediate state observed in FECs exhibiting misfolding, measured with respect to the length of the native state, shows peaks (red) at many of the same positions seen in native unfolding and refolding FECs. The position of the unfolded state is shown as a single bar at 54.7 nm. **e** The distribution of the cumulative unfolded contour lengths of the intermediate states at which misfolding branched from the native folding pathways, measured with respect to the length of the native state

**Table 2 Misfolded states and branching points**

| Misfolded state | Cumulative $\Delta L_c$ measured from native state (nm) | Unfolded $\Delta L_c$ where misfolding branches from native path (nm) |
|---|---|---|
| $M_1$ | $25.0 \pm 0.3$ | $30 \pm 1, 39 \pm 1, 44 \pm 1$ |
| $M_2$ | $20 \pm 1$ | $30 \pm 1, 39 \pm 1, 44 \pm 1$ |
| $M_3$ | $5.0 \pm 0.3$ | $15 \pm 2, 30 \pm 1$ |

The cumulative $\Delta L_c$ measured with respect to the native state (i.e., length of unfolded polypeptide) is shown for each misfolded state, and for each of the intermediates at which a misfolded state branches off from the native folding pathway. All values expressed as mean ± s.e.m.

different denaturant modalities necessarily induce different folding mechanisms[63, 65]. Indeed, the correspondence between the stable-core intermediates observed here and the transition state deduced from phi-value analysis suggests that qualitatively similar mechanisms and barriers are being probed in both the single-molecule and ensemble studies, despite the apparent differences. Moreover, evidence for intermediates was found in some previous studies of apo-SOD1[25, 27, 28, 31], and even studies showing two-state folding suggested the possibility of early unfolding of those strands with weaker inter-strand interactions[18, 30]. Non-cooperative unfolding has also been observed for other beta-barrel proteins with similar structure, such as β2-microglobulin[66].

What is unusual here is thus not the presence but the degree of non-cooperativity: most globular proteins of this size fold via few or no intermediates, whereas we detect at least eight distinct intermediates in the folding of SOD1, with length changes consistent with the sequential addition/removal of individual β-strands in the native structure. There may even be additional intermediates that were not captured because our measurements are only sensitive to structural transitions that change the end-to-end length of the protein. The intermediates do not occur in a deterministic sequence, however: instead, strands tend to form or come apart in groups of variable size (drawn from within the overall sequence of states as in Fig. 3b) that change from one pull to the next. The picture that emerges is thus of an energy landscape that has a large number of relatively small barriers between intermediates (reflecting the near-equilibrium transitions) arranged to create multiple pathways that can be followed during the folding (Fig. 3a). We note that intermediates corresponding to sequential removal of strands have been observed in membrane-bound β-barrels[67], but not yet in globular β-barrels. It is tempting to speculate that such low folding cooperativity may be related to the propensity of SOD1 to misfold, but the correlation between cooperativity and misfolding tendency is unclear, given that other misfolding-prone proteins like PrP can fold quite cooperatively[39, 43].

One notable outcome of the ability to distinguish intermediates differing by the folding of only a single strand is that it allows us to pinpoint where misfolding occurs, by looking at how the transitions into and out of the misfolded states relate to the on-pathway intermediates. In contrast to a protein like PrP, where misfolding was found to occur not via an intermediate but directly from the unfolded state[39, 44], misfolding in SOD1 always involved branching off the pathways for native folding at an intermediate—the intermediates thus acted to mediate

misfolding. Relating the histogram of the branching points identified from the FECs exhibiting misfolding (Fig. 6e) to the intermediates in native folding (Fig. 3b, black), we deduced the points on the native pathways at which the misfolded states $M_1$–$M_3$ could have formed (Fig. 3b, red), identifying several possible misfolding pathways. Remarkably, both native folding and misfolding appear to involve the formation of the same native-like stable core. It is only once this core has formed that misfolding occurs around it.

These results suggest a misfolding mechanism in which the protein retains a partially native core, with the misfolding centered primarily on the two functional loops (loops IV and VII). Such a picture is consistent with previous work finding that these loops tend to become very flexible in reduced apo-SOD1 monomers, capable of forming an aggregation prone interface[16, 55, 58], and that they are a primary locus for misfolding[34, 35]. Furthermore, exposure of the hydrophobic β-barrel core via partial local unfolding has been proposed as a driver for misfolding and aggregation[30]. The detailed window into the misfolding process offered by the single-molecule folding trajectories not only confirms the importance of the functional loops, it also shows directly that specific partially folded intermediates play a crucial role by mediating between the native and non-native folding pathways. Such intermediates may provide effective targets for pharmacological chaperones that prevent misfolding[45].

The picture of misfolding presented here suggests a tantalizing explanation for the tendency of SOD1 to misfold and its unusual ability to undergo prion-like conversion: it is fairly easy to reach the misfolded states from the native fold, simply by reconfiguring the least-stable part of the structure. Future studies of SOD1 in the single-molecule regime hold the promise to probe more directly the interactions between misfolded and native SOD1 that drive conversion, relating the behavior of isolated apo-SOD1 monomers to propagated misfolding, as well as to investigate ligands that prevent misfolding and aggregation[68], identifying their molecular mechanisms of action[45].

## Methods

**Protein synthesis and purification.** Monomeric human Cu,Zn-superoxide dismutase (SOD1) was engineered to contain flanking cysteine residues at either terminus for DNA handle attachment. The solvent-exposed sulfhydryls were removed by amino acid substitutions (C6A/C111S), to prevent incorrect handle attachment, and a second pair of substitutions (F50E/G51E) was made to prevent dimerization of the resulting protein[56]. The open reading frame was synthesized (DNA2.0) and inserted into the plasmid pJExpress401 to produce an N-terminally tagged His$_6$ fusion protein (sequence listed in Supplementary Table 2). This plasmid was used to express the protein in BL21(DE3) E. coli (EMD Millipore), adding 50 μM ZnSO$_4$ and CuSO$_4$ to the growth medium (LB with 35 μg ml$^{-1}$ kanamycin) before overexpression was induced with 1 mM IPTG. After growing cells for 16 h at 37 °C, they were harvested and resuspended in 50 mM Tris-Cl (pH 7.5), 500 mM NaCl, 25 mM imidazole, 1 mM ZnSO$_4$, 1 mM CuSO$_4$, 250 μg hen egg white lysozyme, one EDTA-free protease inhibitor cocktail tablet (Roche), and ten units DNase I. Resuspended cells were frozen in liquid N$_2$ and stored at −80 °C prior to purification.

For purification, samples were thawed, sonicated, and centrifuged at 28,000 × g for 1 h. The supernatant was loaded onto a 5 ml Ni$^{++}$-charged immobilized metal affinity column that had been equilibrated with 50 mM Tris-Cl (pH 7.5), 50 mM NaCl, 10 mM MgCl$_2$, 1 mM CaCl$_2$, and 25 mM imidazole. Non-binding proteins were washed thoroughly using equilibration buffer, then bound protein was eluted with 50 mM Tris-Cl, 50 mM NaCl, 10 mM MgCl$_2$, 1 mM CaCl$_2$, and 600 mM imidazole. Eluted fractions were analyzed using 15% SDS-PAGE; fractions containing monomeric SOD1 were then pooled and dialyzed into 25 mM KH$_2$PO$_4$/K$_2$HPO$_4$ buffer, pH 7.0.

**SOD1 activity assay.** To test that the SOD1 samples were properly folded and fully active, enzymatic activity was measured following an assay reported previously[69]. Briefly, xanthine oxidase was used as a source of superoxide radicals, which react with the colorimetric agent WST-1 to produce colored WST-1 formazan (absorbs light at 450 nm). SOD1 converts superoxide into peroxide and water and thereby scavenges superoxide to reduce the rate of formation of WST-1 formazan. The inhibition of the reaction between WST-1 and superoxide by SOD1

was quantified by comparing the rate of formation of WST-1 formazan in the absence and presence of SOD1 and normalizing by the rate in the absence of SOD1 to determine the percent inhibition.

**Sample preparation.** DNA handles were attached to the protein as described previously[40]. Briefly, SOD1 was reduced with tris(2-carboxyethyl)phosphine (TCEP). After removal of excess TCEP by four-fold spin filtration (Amicon Ultracel-10K 0.5-ml centrifugal filters), SOD1 was activated with 2,2′-dithiodipyridine (DTDP), incubating the protein in 100-fold molar excess of DTDP at 4 °C for 24 h. DTDP-activated SOD1 was purified from excess reagent by spin filtration as done for TCEP, and then reacted with sulfhydryl-labeled DNA handles prepared by PCR (one 798 bp, labeled by biotin, the other 2113 bp, labeled with digoxigenin), incubating equal concentrations of protein and handles at 4 °C for 24 h. The labeling stoichiometry and efficiency were assessed by mass spectrometry, finding that SOD1 was labeled with precisely two DTDP molecules at ~30% efficiency. We also used mass spectrometry to verify that the functionalization was specific to the terminal cysteines, finding that C6A/C111S SOD1 (which contains only internal cysteines) did not react with DTDP, in contrast to the monomeric construct with terminal cysteines. The internal cysteines in SOD1 were thus non-reactive. We verified that the SOD1 monomer retained its enzymatic activity after labeling with DTDP (Supplementary Fig. 5), implying that the functional fold is preserved in the protein–DNA construct.

SOD1–DNA constructs were incubated at ~100 pM with 250 pM polystyrene beads (600-nm diameter labeled with avidin, 820 nm-diameter labeled with antidigoxigenin), to form dumbbells. Dumbbells were diluted to ~500 fM in 50 mM MOPS, pH 7.0, with 200 mM KCl and oxygen scavenging system (8 mU μL$^{-1}$ glucose oxidase, 20 mU μL$^{-1}$ catalase, 0.01% wt/vol D-glucose), before insertion into a sample cell for the optical trap. In order to chelate metal ions and keep the protein in the apo state, the final dilution buffer contained 2 mM EDTA. Note that any residual metal ions that might have remained bound to folded SOD1 would be chelated by the EDTA in the measuring buffer once the protein was first unfolded, ensuring that the protein was in the apo state during SMFS measurements.

**FEC measurement and analysis.** Samples were measured using a home-built optical tweezers apparatus with two independently controlled traps described previously[40]. Briefly, a 5 W, 1064-nm laser was used to create two traps with orthogonal linear polarization. The trap position was controlled independently in each axis of each trap by acousto-optic deflectors, which were also used to modulate the trap stiffnesses. The positions of the two beads within their respective traps were measured using position-sensitive diodes to collect the light from a 833-nm detector laser that was scattered by the beads. FECs were measured by moving the two traps apart (for unfolding) or together (for refolding) in 1–2 nm steps at a pulling rate of 200 nm s$^{-1}$, sampling data at 20 kHz while using an eight-pole Bessel filter (Krohn-Hite) set to the Nyquist frequency for online anti-alias filtering. Data within each step were then averaged. The stiffnesses of the two traps were calibrated as described previously[40] and set to 0.37 and 0.54 pN nm$^{-1}$. The waiting time at near-zero force between successive pulls was varied between 0.5 s (2390 FECs from ten molecules), 5.5 s (327 FECs from two molecules), and 10.5 s (333 FECs from two molecules).

To determine the change in the contour length ($\Delta L_c$) that occurred during unfolding and refolding transitions, individual FECs were fit to an extensible WLC model[57]:

$$F(x) = \frac{k_B T}{L_p}\left[\frac{1}{4}\left(1 - \frac{x}{L_c} + \frac{F}{K}\right)^{-2} - \frac{1}{4} + \frac{x}{L_c} - \frac{F}{K}\right], \qquad (1)$$

where $L_p$ is the polymer persistence length, $L_c$ is the contour length, and $K$ is the elastic modulus. We used two WLCs in series, one to account for the handles and the other to account for any unfolded protein that was present[39, 40]. The parameters for the handle ($L_c$, $L_p$, and $K$) were allowed to vary freely to fit the branch of the FECs representing the folded state (with the unfolded protein contour length set to zero), but they were subsequently treated as fixed parameters for fitting the intermediate and unfolded branches of the FECs. The protein $L_p$ and $K$ values were also treated as fixed parameters, with $L_p = 0.85$ nm and $K = 2000$ pN[40]. Hence the fits for the intermediate and unfolded states involved only a single free parameter, the contour length of unfolded protein present in those states. We determined $\Delta L_c$ for individual unfolding transitions by fitting each side of every identifiable rip in the FECs to WLCs with different lengths of unfolded protein. Fitting was done manually, analyzing only intermediate states that persisted for at least 10–15 ms and that displayed "rips" with force and extension changes larger than the fluctuations in measurements of DNA handles only (Supplementary Fig. 6) at comparable forces.

The contour length change ($\Delta L_c$) was converted into the number of amino acids unfolded, $n_U$, via $n_U = (\Delta L_c + d_T)/L_c^{aa}$, where $d_T$ is the distance between the termini and $L_c^{aa} = 0.36$ nm aa$^{-1}$ is the crystallographic contour length of an amino acid[40]. For apo-SOD1$^{2SH}$, we measured $d_T = 0.9$ nm from the crystal structure[58]. Using these values, complete unfolding of apo-SOD1$^{2SH}$ yielded $n_U = 154 \pm 1$, matching the 153 aa in the native structure of the protein. Note that the total contour length change upon unfolding was used as an additional check that the DNA handles were

attached to the terminal cysteines, as otherwise the full length change corresponding to 153 aa would not be observed.

Analysis of the intermediates was based on the cumulative $\Delta L_c$ measured from the folded state, rather than the incremental $\Delta L_c$ for individual transitions, because studying the cumulative $\Delta L_c$ is more informative when many intermediates are present[59]. The cumulative $\Delta L_c$ allows not only the length changes between states to be captured but also the order in which they occur, and it permits easier comparison of FECs when each curve may contain a different number of intermediates and a different set of states. The distributions of cumulative $\Delta L_c$ were similar for all molecules, hence they were combined into a single histogram.

The peaks in the contour-length histograms were fit with Gaussians under the assumption of normally distributed errors in the WLC analysis. Because the folded and unfolded branches of the FECs were the most well-defined and thus easiest to fit, the peak for the total contour length change from the folded to unfolded state was narrower than all other peaks (half-width ~1 nm). WLC fitting of the intermediate states was more error-prone because the intermediates were generally short-lived, leading to wider peaks in the contour length histograms (half-width ~2 nm). When comparing the measured $\Delta L_c$ to the values expected from the structural models, we assumed an error of 0.3 nm in the position of each terminus (estimated from the variance in NMR and crystal structures[16, 58]), leading to an estimated error in the expected values of 0.3 nm.

**Folding pathway analysis.** To create transition maps for native folding (Fig. 3a), every FEC displaying a total $\Delta L_c$ consistent with natively structured SOD1 was divided into sections assigned to a particular structural state (N—native, U—unfolded, $I_1$ to $I_8$—the intermediates between U and N) based on the cumulative $\Delta L_c$ measured with respect to the native state. The midpoints between the peaks in the cumulative $\Delta L_c$ distributions (Figs. 1g, 2b) were used as thresholds delineating each intermediate: segments of FECs with cumulative $\Delta L_c$ between the thresholds around a particular intermediate were assigned to that state. The state occupied at the highest forces was assigned to U. Pairwise transitions between states were then tabulated from all FECs.

To deduce possible structures for the intermediate states, all possible substructures derived from the native fold of SOD1 were enumerated, subject to several constraints. (1) Loops likely do not provide mechanical resistance, hence the intermediates were restricted to combinations of folded and unfolded strands (e.g., β1–7 folded with β8 unfolded, or β1–3 and β7–8 folded with β4–6 unfolded, etc.). (2) Intermediates that left the end-to-end extension of the molecule unchanged cannot be detected and hence were excluded. (3) Strand combinations with few or no tertiary contacts in the native structure were excluded because they would provide no mechanical resistance. (4) Simultaneous unfolding and refolding of unconnected strands was excluded, on the assumption that each event would be observed individually. (5) Strands were assumed to retain native-like structure within each substructure. The expected cumulative $\Delta L_c$ relative to the native state was then calculated for each structure (Supplementary Table 1), using $\Delta L_c = 0.36 \times n_U + \sum_j (d_T)_j - (d_T)_0$, where $n_U$ is the total number of unfolded amino acids, $(d_T)_0$ is the distance between the termini (i.e., the end-to-end distance) in the native structure, and $(d_T)_j$ is the end-to-end distance of the structured region $j$ in the intermediate. Uncertainties in the cumulative $\Delta L_c$ predictions were estimated from the variance in the crystal and NMR structures of SOD1[16, 58].

For each intermediate in a FEC, the measured $\Delta L_c$ was compared to the predicted values for all putative intermediate structures to identify potential matches, defined as any substructures where the predicted and measured $\Delta L_c$ differed by < 4 nm (to allow for error in the state assignment). This comparison thus explicitly allowed for the possibility that a given peak in the cumulative $\Delta L_c$ distribution ($I_1$–$I_8$) could represent more than one distinct structure. The substructures consistent with the states in the FEC were then used as nodes in a network analysis to find all possible pathways consistent with the data in that curve (Supplementary Fig. 1). The analysis was repeated for each FEC to obtain the set of pathways consistent with all the data. Based on studies showing that β8 is relatively unstable[18, 30], the results were further refined to exclude β7–8 as a stable substructure. Additionally, because the errors in state assignment should be random, we excluded any substructure that was systematically on one edge of the range permitted for matching predictions and observations (with a mean difference between the lengths of more than twice the standard deviation in the putatively matching intermediate lengths); the only state affected was β2–8. This analysis was used to identify the minimal set of pathways able to explain the data, summarized in Fig. 3b. Note that other pathways were also found, but they were consistent with much less of the data; some examples of such pathways are illustrated in Supplementary Fig. 2.

**Computational simulations.** Simulations were performed following the method described by Habibi et al[61]. Briefly, we initialized the simulations with the structure of the monomer as deduced from the human SOD1 crystal structure[58] (PDB ID: 3ECU), removing the disulfide bridge and ions to conform to the experimental conditions (reduced apo-SOD1). The initial model was equilibrated via explicit-solvent all-atom MD simulations before converting the all-atom model to a coarse-grained, Cα-Go-type model. We first validated our implementation of the method by replicating the results found by Habibi et al. on a loop-deletion mutant of SOD1, using similar parameters to simulate pulling experiments (pulling speed 1 m s$^{-1}$,

spring constant 6 pN nm$^{-1}$). We then applied the method to the full SOD1 monomer, simulating 26 unfolding trajectories. Intermediates in the unfolding trajectories were identified by manual inspection as structures that persisted longer than 10 ps in a single trajectory. For additional details, see Supplementary Methods.

**Truncation mutants.** Samples of the truncation mutants β1–3 and β2–3 were synthesized as peptides (CanPeptide Inc.) consisting of residues 1–39 and 9–40, respectively, with a Cys residue added to each terminus for attaching DNA handles. Peptides were functionalized with DTDP and attached to DNA handles as described above. SMFS measurements were performed and analyzed as for SOD1. The β1–3 peptide aggregated too readily to attach DNA handles, however, and thus could not be studied. CD spectra were measured to confirm the β-structure of the peptide, using a Jasco J-810 spectropolarimeter with a peptide concentration of 20 μM in 10 mM sodium phosphate pH 7.0 over a path length of 1 mm. Background spectra of the buffer only were subtracted and the sample spectra converted to units of mean residue ellipticity. The secondary structure content of the peptide was analyzed from the spectrum using BeStSel[70].

**Code availability.** Code is available from the corresponding author upon reasonable request.

**Data availability.** All data that support the findings of this study are available from the corresponding author upon reasonable request.

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

## Acknowledgements

This work was funded by the Alberta Prion Research Institute, Alberta Innovates Technology Futures, and the National Research Council of Canada. S.S.M., D.R.D., and U.A. were supported by fellowships from Alberta Innovates Health Solutions.

## Author contributions

M.T.W., S.S.M., and D.R.D. designed the research; C.G. and D.R.D. designed the protein constructs; C.G. provided reagents; S.S.M. prepared sample and performed experiments; S.S.M., Z.S., L.R., U.A. analyzed the data; Z.S. performed and analyzed simulations; S.S. M., Z.S., C.G., and M.T.W. wrote the manuscript; all authors edited the manuscript.

## Additional information

**Competing interests:** The authors declare no competing financial interests.

