## [Peer Review File · Nature Communications]

Reviewers' comments:

Reviewer #1 (Remarks to the Author):

In this manuscript, Mojumdar et al used optical trapping to investigate the mechanical unfolding, folding and misfolding of a monomeric mutant of SOD1. They found that SOD1 unfolds via multiple intermediate states, and refolds via multiple (and possibly similar) intermediate states. Based on the predicted ΔL_c , the authors concluded that SOD1 folds/unfolds following a hierarchical pathway involving the sequential formation/disruption of each β -strand in the native structure. The results presented here are very interesting, and could provide a novel perspective on the folding/unfolding mechanism of SOD1. The authors should address the following issues before the manuscript becomes suitable for publication.

- 1) The data analysis is quite confusing. Typically, ΔL_c for each individual intermediate state is reported, not the cumulative ones. Reporting cumulative ΔL_c may lead to wrong grouping of the events. I suspect that the wide ΔL_c distribution for some intermediate states is simply due to the grouping of different and unrelated intermediate states into one population.
- 2) Fig. 3 on misfolded state: are these states true misfolded states? What is the life time of such misfolded states? Can they be relatively long-lived intermediate state?
- 3) The data was compiled from more than 2000 FEC. From how many independent molecules did the authors obtain these 2000+ FEC? Are ΔL_c histograms from these different molecules similar?
- 4) The conclusion that the unfolding of SOD1 corresponds to the peeling of individual beta strands need more experimental evidence. Due to the degeneracy of the ΔL_c from different structures, it is necessary to verify such a claim using additional protein constructs.
- 5) Fig. S1, the activity of labeled SOD1: the authors need to provide more information about the samples. How are the labeled SOD1 prepared and purified? What is their purity?

Reviewer #2 (Remarks to the Author):

Misfolding of Cu/Zn-superoxide dismutase (SOD1) triggered by amino acid substitutions has been proposed as a major pathological process in amyotrophic lateral sclerosis. Many researchers have attempted so far to understand the folding process of SOD1 proteins with various experimental/theoretical methods. In this paper, the authors have characterized the folding and misfolding of SOD1 proteins with single-molecule force spectroscopy. They artificially extended a single SOD1 molecule by using dual-trap tweezers, and the force-extension curves of SOD1 unfolding/refolding were analyzed. The curves were found to be interrupted by "rips", based upon which the authors proposed the folding intermediates of SOD1.

The paper appears to be interesting but cannot be recommended for publication because of the following reasons.

- 1: The results in this paper do not describe (or actually, do not have any relation with) the prion-like activity of misfolded SOD1.
- 2: While the authors described that single-molecule methods have not been applied to study misfolding in SOD1, Solsona et al. already examined SOD1 folding process with single-molecule force

clamp spectroscopy (J Biol Chem 2014 289 26722).

3: The "tweezers" are attached to the N- and C-termini of SOD1; therefore, in this experiment, the site from which the unfolding starts is limited to the N- and C-termini. It is possible that SOD1 starts to unfold at the site(s) other than the N- and C-termini, but such unfolding process cannot be evaluated by the method in this study. Also similarly, the N- and C-termini of SOD1 would interact with each other during the early process of SOD1 refolding. The experimental system in this study is therefore supposed to be quite artificial. The authors should examine several permutants of SOD1. However, I think that the pathologically relevant misfolded conformation(s) cannot be revealed by the experimental methods in this study.

4: The authors described that SOD1 folding/refolding are characterized by the high degree of non-cooperativity; however, the high degree of non-cooperativity appears to be artificial due to the experimental method, single-molecule force clamp spectroscopy.

5: The preparation of SOD1 samples was not clearly described. Both metallation and thiol-disulfide status of SOD1 (or actually, artificially monomerized SOD1) were not experimentally confirmed. Also, SOD1 in this study was reduced with TCEP, so the intramolecular disulfide bond was supposed to be reduced. It is quite obscure where in SOD1 the sulfhydryl-labeled DNA handles were attached.

Reviewer #3 (Remarks to the Author):

This manuscript by Mojumdar et al. uses high-resolution single molecule optical trap spectroscopy to monitor the unfolding and refolding of demetalated, monomeric SOD. Unfolding and refolding are monitored by force ramp experiments and examination/analysis of the force extension curves. The authors find a very non-cooperative multi-step unfolding and refolding curve. Using Occam's razor to interpret these 'rips' or changes in extension, the authors propose a stepwise mechanism of unfolding and refolding for this protein. They also find evidence for misfolding from some of these intermediates, which together with their structural interpretation suggest regions to target in the misfolding trajectory of this medically important protein.

Overall, I like this manuscript and think it is of interest to a broad audience. I am somewhat concerned that it does not present very much data (just FECs on the one protein) and might be more impactful if it made some mutations to test the structural pathway it proposes. I assume those experiments will be forthcoming and given the general interest in the unfolding of SOD, do think it represents an important advancement. I do have a few question/concerns that should be addressed.

1. Data analysis/interpretation:

a. How were the rips identified in the force ramp experiments? Was this done manually or by a nonbiased algorithm? The representative curves seem to show some noise on the level of the size of the rips – are those very transient 'equilibrium' visitations to the intermediate?

b. The author's explanation of their data is based on a very native 'GO-like' model – ie the unfolding of individual strands off the edges of the beta sheet. But isn't this the only kind of intermediate one could see with this method (regions coming off from the edges)? If an intermediate formed by unfolding an internal beta strand or a hairpin, there would be no change in extension – so how would this be observed? Is it possible that there are many more intermediates not seen in these FEC curves?

c. I am confused by the fact that the same intermediates are seen in both folding and refolding? For example, in Figure 1 c,d and e the last unfolding rips is just under 10pN and the first refolding rip is

also around that force. But these are not equilibrium experiments – shouldn't refolding occur at a lower force than unfolding? Perhaps this is just a result of showing representative curves? Shouldn't there need to be a barrier separating the two in order to transiently form an intermediate – I would assume that you would not see all the same barriers in each direction. Since the authors are proposing a very ordered optional set of steps, perhaps they could put them on some energy (or even force) diagram to help me (and others) understand.

d. I think the authors could explain more clearly how they know the misfolds are distinct from the intermediate states. The contour lengths presented in table 2 for M1, M2 and M3 are similar to the contour length of these intermediates. I'm sure the authors have thought about this, but more discussion in the paper of how they know the misfolded states are distinct from the intermediate states would be useful.

e. On page 8-9, they compare their results to ensemble studies. Please clarify if these ensemble studies are also on an un-metalated monomer.

2. Sample prep:

a. How do they know that the protein is effectively demetalated? Is it already known that simple addition of chelators is enough? Would the authors get the same result by first unfolding to high force (and in essence removing the bound metals) and then recording future ramps (or is the metalated form too stable to unfold at the accessible forces?)

b. The ability to attach DNA handles to a protein with several 'internal' cysteines seems novel and important. The authors claim to mutate the surface exposed cysteines and, by using mass spec, they can determine that the handles are only interacting with the engineered terminal cysteines. I am not exactly sure how they did this (I imagine, they showed dttp addition (by mass spec) on constructs without the engineered cysteines and no DTDP on those without the engineered cysteines). This should be explained more clearly and perhaps added to supplementary material.

c. I suggest adding information about the protein – that is is human SOD with some mutations to remove external cysteines (defined by what? I assume solvent exposed surface area) and prevent dimerization. This information can be found in the methods – but it is something a casual reader would want to know.

Response to Referee Comments

Reviewer #1:

In this manuscript, Mojumdar et al used optical trapping to investigate the mechanical unfolding, folding and misfolding of a monomeric mutant of SOD1. They found that SOD1 unfolds via multiple intermediate states, and refolds via multiple (and possibly similar) intermediate states. Based on the predicted ΔL_c , the authors concluded that SOD1 folds/unfolds following a hierarchical pathway involving the sequential formation/disruption of each β -strand in the native structure. The results presented here are very interesting, and could provide a novel perspective on the folding/unfolding mechanism of SOD1.

We thank the referee for these positive comments.

The authors should address the following issues before the manuscript becomes suitable for publication.

1) The data analysis is quite confusing. Typically, ΔL_c for each individual intermediate state is reported, not the cumulative ones. Reporting cumulative ΔL_c may lead to wrong grouping of the events. I suspect that the wide ΔL_c distribution for some intermediate states is simply due to the grouping of different and unrelated intermediate states into one population.

We apologize that we did not explain the motivation for presenting the data in terms of cumulative contour lengths (as opposed to incremental contour length changes). The referee is correct that in many papers, only the incremental ΔL_c for each transition (*i.e.*, the change of contour length between successive states) is used. This approach works well when the folding is two-state, or when only a few intermediates are present, but it becomes less informative when many intermediates are present. There are a couple of disadvantages that arise from using the incremental ΔL_c :

- 1) There is important information contained not only in the value of each incremental ΔL_c but also in the order of these length changes. This information about the ordering of the length changes is discarded when the analysis is confined to the incremental ΔL_c . In contrast, by analyzing the length changes in terms of the cumulative contour length of unfolded protein, one can determine when in the unfolding/refolding sequence a transition with a given incremental ΔL_c occurred. Indeed, in our view, reporting incremental ΔL_c only is more likely to lead to erroneous groupings of intermediate states. As a concrete example, consider an unfolding transition that contains two sequential intermediates, the first with an incremental ΔL_c of 8 nm from the folded state, the second with an incremental ΔL_c of 10 nm from intermediate 1, and a final incremental ΔL_c to the unfolded state of 15 nm, for a total cumulative ΔL_c of 33 nm from folded to unfolded. Assume that there is a measurement error of 2 nm for fitting the individual FECs (*i.e.*, the distribution of ΔL_c values is given by a Gaussian with width 2 nm). If the incremental ΔL_c is plotted, then as seen

Figure R1: Contour length analysis. (A) Only 2 peaks are seen when plotting incremental ΔL_c (one of the peaks being broader). (B) 3 peaks are seen using the cumulative ΔL_c for a protein with intermediates with incremental ΔL_c of 8, 10, and 15 nm.

in Fig R1A the two intermediate states appear to be a single peak with wider distribution; even if one knows that there are two peaks present, it's difficult to resolve them, and the plot does not show the order in which changes occur. If the cumulative ΔL_c is plotted, in contrast (Fig R1B), then the intermediates are easily resolved, and the sequence of unfolding transitions (and their lengths) is readily apparent from the plot.

- 2) A second problem is that one can't always detect the molecule passing through every intermediate in every FEC, owing to a combination of factors (discussed in more detail below, see point 4). As just one example, the exponential distribution of state lifetimes leads to the "missing event" problem, where some of the transient visits to intermediate states are simply not detected because they fall below the detection threshold in terms of duration (in our work, a state must persist for at least 10–15 ms to be included in the analysis). If each intermediate could be detected with assurance in every FEC, then one could easily make a distribution of the incremental ΔL_c from the folded state to the first intermediate, another for ΔL_c between the first and second intermediates, and so on to the unfolded state, solving the problem of the order of events discussed above. (As an aside, note that simply adding each of these distributions together, incrementing ΔL_c from one intermediate to the next, would then just replicate our cumulative ΔL_c distribution.) Unfortunately, we can't do anything like that here: we observe different subsets of the population of intermediates in any given FEC. The variable number of intermediates likely reflects not just missed transitions (most of the intermediates do indeed have quite short lifetimes in the FECs, as seen by not infrequent rapid quasi-equilibrium hopping, and thus would expect to give rise to missed events), but also multiple pathways through the relatively low barriers separating the intermediates (where we know that the barriers must be fairly low because of the short lifetimes).

In contrast, using the cumulative ΔL_c brings a couple of advantages that are important in the present work:

- 1) It makes it easier to compare FECs containing different numbers of intermediates, sidestepping some of the problems created by the sampling of only a subset of all possible intermediates in any given FEC. Suppose we label the folded state as length 0 and the unfolded state as length 1, and we consider a protein with three short-lived intermediates having lengths 1/4, 1/2, and 3/4. If in one FEC the intermediates at 1/2 and 3/4 are observed but 1/4 is missing, in the next FEC the ones at 1/4 and 1/2 are the only ones observed, and another FEC shows only 1/4 and 3/4, then creating the cumulative ΔL_c distribution will show three peaks, one corresponding to each of the three intermediates that are detected at various times. The total amount of unfolded vs folded polypeptide in each intermediate can also be easily read off the distribution. In contrast, plotting the distributions of incremental ΔL_c in the first, second, and third transitions would show only two peaks in each distribution: 1/4 and 1/2. Interpreting these lengths in terms of the total amount of folded vs unfolded structure then becomes much less straightforward (it involves a conditional analysis of each graph, not something that can easily be done by visual inspection). The cumulative ΔL_c thus provides the simplest visual representation of the full set of intermediates observed.
- 2) Using the cumulative ΔL_c makes it easier to identify FECs containing misfolding, where the total ΔL_c does not match the value for the native fold (see more detailed discussion below in point 2 for how we define misfolding), because one focuses on the total change in length (which has lower uncertainty than ΔL_c for transient intermediates, as discussed in the methods section). Furthermore, the cumulative ΔL_c makes it easier to find the branching points where

misfolding diverges from native folding, by comparing the FECs containing misfolding to those without: the last intermediate state on the native folding pathways that is observed before a non-native length is seen is readily identified from the cumulative ΔL_c but more difficult to determine from incremental ΔL_c (again, it would require some sort of conditional analysis of the graphs, something considerably more complex than simple visual inspection).

We note that we are certainly not the first to analyze unfolding using the cumulative ΔL_c rather than the incremental ΔL_c : it has been used before in the force-spectroscopy literature to improve the analysis of FECs containing multiple intermediates. See for example the work in Bosshart *et al* (2012) *Biophys J* 102:2202-2211 and Yu *et al* (2017) *Science* 335:945-950, where the authors had to contend with many intermediates. As in this previous work on multi-state systems, we feel that the cumulative ΔL_c is the most effective choice for analyzing the data.

Regarding the widths of the peaks in the ΔL_c distribution, from our previous work we found that fitting long-lived states (*e.g.*, as in a two-state system where each of the states lasts long enough to produce a segment of the FEC where the shape of the curve is well-defined) to the WLC model resulted in a width of about 2 nm (Neupane *et al* (2014) *PLoS One* 9:e86495). The width of the peak for the fully unfolded state matches this number, which makes sense because the unfolded state persists for a long time in the measurement. The intermediate states, however, are short-lived and quasi-equilibrium, so that the molecule does not tend to persist very long in any given intermediate. As a result, only quite short sections of the FECs are available for fitting and the fitting errors are thus larger, leading to larger peak widths for the intermediates.

We have revised the Methods section to motivate the use of the cumulative contour length change in our analysis.

2) *Fig. 3 on misfolded state: are these states true misfolded states? What is the life time of such misfolded states? Can they be relatively long-lived intermediate state?*

The referee's question ultimately relates to how one defines a misfolded state. We are effectively using a definition based on the widely-used 'energy-landscape' picture of protein folding, wherein the native structure is the most thermodynamically stable. (At least this is the situation for isolated monomers, what we have in our experiments—when monomers are at sufficiently high concentration that they interact strongly, the landscape may change so that misfolded aggregates like amyloids are more stable—see, for example, Eichner & Radford (2011) *Mol. Cell* 43:8–18.) In this case, a misfolded state is one that is a metastable kinetic trap off of the "regular" pathway(s) leading to the native state, such that if the misfolded state is occupied, it prevents the protein from forming the native structure as fast as it would normally do. As such, it is indeed a type of intermediate state (as effectively required by the landscape picture, which posits the native state as being the most stable), but it differs from 'on-pathway' intermediates that lead directly to the native state because it is 'non-productive', separated from the native state by energy barriers that are sufficiently high so as to generate a trap. The protein either has to surmount the high barriers, or backtrack to return to the regular native pathway. This definition of misfolding effectively matches the definitions used by others in the field, see for example Jahn & Radford (2008) *Arch Biochem Biophys* 469:100-117, or Borgia *et al.* (2010) *Nature* 474:662–665 and (2015) *Nat. Commun.* 6:8861.

As seen in our data, in the vast majority of cases SOD1 folds rapidly via a sequence of short-lived intermediates into the native structure. The states we identify as misfolded are the ones

where the protein becomes trapped in a structure that is not the native structure. These states are indeed distinct from the partially folded intermediates on the regular pathway to the native state: their lengths differ (in most cases) from the lengths of the on-pathway intermediates, and their lifetimes are much longer. Their properties thus match what one would expect for a misfolded state.

To answer the referee's question about the lifetime of the misfolded states, we have added new measurements of the timescale on which the misfolded states relax into the native state, by measuring the fraction of pulls which start from a non-native state as a function of the waiting time between pulls. Given that the native state is the most stable, one would expect the fraction of misfolding observed to decrease with increasing waiting time (as was indeed seen in previous work, *e.g.* Borgia *et al. Nat Commun* 2015). We found that the misfolded fraction decreased with a rate constant of approximately 0.17 s^{-1} , representing an average lifetime for the misfolded states of 6 s. This lifetime is very long compared to the normal folding process, which is completed on a timescale of less than 0.5 s (the time to go from $\sim 10 \text{ pN}$ to $\sim 0 \text{ pN}$).

We have revised the manuscript in the Results section to explain more clearly what we mean by misfolded states and to present the new lifetime data.

3) The data was compiled from more than 2000 FEC. From how many independent molecules did the authors obtain these 2000+ FEC? Are deltaLc histograms from these different molecules similar?

The data were measured from 6 different molecules, with an average of about 400 pulls each. This is enough data from each molecule to check that the histograms from different molecules are indeed similar. A statement to this effect has been added to the revised manuscript in the Methods section.

4) The conclusion that the unfolding of SOD1 corresponds to the peeling of individual beta strands need more experimental evidence. Due to the degeneracy of the deltaLc from different structures, it is necessary to verify such a claim using additional protein constructs.

We agree with the referee that it would be helpful to have additional experimental evidence to support the model of the folding that we propose. The properties of SOD1 folding, however, make this a very challenging task, for a number of reasons:

- SOD1 is unusually sensitive to mutations, reflected in the fact that over 150 point mutations perturb the protein sufficiently so as to be linked to familial ALS (Pasinelli & Brown (2006) *Nat. Rev. Neurosci.* 7:710–723).
- Not only is apo-SOD1 not very stable thermodynamically, it also has a large number of intermediates that are close in stability (all unfolding within a force range of less than 10 pN) and it is aggregation-prone.

These properties combine to make it difficult to apply most of the approaches typically used for identifying the nature of transient intermediate states. Note that it is very challenging to use ensemble methods for such a purpose: even leaving aside the fact that such studies have generally had difficulty detecting intermediates in SOD1, there are too many transient states present, their lifetimes are too short, and their occupancy cannot be synchronized, making it impractical to study them in an ensemble. Hence we are limited to single-molecule approaches.

One strategy that has been used quite successfully on other proteins is to make truncation mutants that isolate the putative intermediates, as in the elegant work by Rief and colleagues on calmodulin (Stigler *et al.* (2011) *Science* 334:512–516) and Hsp90 (Jahn *et al.* (2016) *PNAS* 113:1232–1237). However, we quickly run into problems when trying to apply this approach to SOD1. For example, the very first construct we would want to make in order to verify the initial step in our proposed model would involve a simple truncation of the C terminus (strand 8 and loop 7). Such a truncation, however, generates a form of the protein that is exceptionally difficult to study because of aggressive aggregation. Indeed, the G127X truncation mutant of SOD1 (in which strand 8 and most of loop VII are removed) not only doesn't fold—instead aggregating rapidly—it is linked to a particularly aggressive form of familial ALS (Jonsson *et al.* (2004) *Brain J. Neurol.* 127:73–88). It might be possible through trial and error to discover some truncation mutants that are experimentally tractable, but such a search would be very laborious and time-intensive, with no guarantee of success.

Nevertheless, we attempted to apply this approach, focusing strategically on validating the “stable core” intermediates in our model. Our reasoning was that this stable core appears to fold reliably and independently, hence a truncation mutant consisting of just the stable core might have a decent chance of being sufficiently well-behaved so as to permit SMFS measurements. We made two constructs, one consisting of the β -sheet formed by strands 1 through 3 (corresponding to the state I_2), and one consisting of the β -hairpin formed by strands 2 and 3 (corresponding to the state I_1). As we had worried, the first construct aggregated too much for us to measure it (not surprisingly, since a significant part of the hydrophobic core of SOD1 is exposed on one side of this sheet). The β -hairpin, however, was sufficiently soluble that we could prepare samples of it for measurement. We were able to confirm that not only did it fold into a stable hairpin, but the rips in the FECs looked very similar to those in the very last intermediate to unfold (and first to refold) in SOD1, as seen below in Fig R2, comparing unfolding curves from the peptide and the last intermediate of SOD1. (We note that the fact that this hairpin is stable on its own is somewhat unusual— β -hairpins isolated from their structural context are typically unstable.) These measurements thus validate both the notion that a small part of the full structure can remain independently stable after everything else has unfolded, as well as the identity of the strand 2–3 hairpin intermediate in the stable core. Furthermore, this

Figure R2: Comparison of hairpin 2–3 peptide and final unfolding transition in SOD1. The hairpin formed by strands 2 and 3 folds independently when made as a peptide. FECs of the hairpin (left) show transitions very similar to the highest-force transitions seen with SOD1 (right), validating the idea that hairpin 2–3 unfolds last.

intermediate places an important constraint on the models of folding that are possible: anything in which strands 2 or 3 unfold early is ruled out.

We also investigated a second strategy for validating the model of the folding, one that originally seemed more promising than truncation mutants: using loop deletions or insertions to change the lengths associated with unfolding specific parts of the protein. This approach would allow the structural features associated with particular intermediates to be identified by seeing which unfolding lengths changed upon changing a particular loop. Such a strategy has been used effectively in the past by

various groups performing AFM measurements of protein unfolding, using either loop insertion/deletion or even domain insertion (see, for example, Carrion-Vazquez *et al.* (1999) *PNAS* 96:11288–11292, Bertz & Rief (2008) *J. Mol. Biol.* 378:447–458, Peng & Li (2009) *J. Am. Chem. Soc.* 131:14050–14056, Scholl *et al.* (2017) *Biophys. J.* 112:1829–1840). As an added advantage, loop deletion mutants of SOD1 have been studied previously and are known to fold reliably (Danielsson *et al.* (2011) *J. Biol. Chem.* 286:33070–33083).

Upon closer consideration, however, this strategy, too, suffers from a fatal flaw: the problem is that the intermediate states in SOD1 do not differ much in stability. Because loops generally have a destabilizing effect owing to the loop entropy (as demonstrated specifically in the context of SMFS by Li *et al.* (2008) *J. Mol. Biol.* 379:871–880 and Li *et al.* (2014) *Angew. Chem.* 53:13429–13433), deleting a loop (*e.g.* loops IV or VII in SOD1) is likely to increase the stability of the neighboring strands. This change in stability is then very likely to change the order in which the intermediates occur, scrambling attempts to identify which intermediate is which. In fact, ensemble studies showed that the change in stability in apo-SOD1 upon loop deletion is significant: deleting loops IV and VII more than doubles the stability (Danielsson *et al.* *J. Biol. Chem.* 2011). Furthermore, removing the loops may also make the protein more cooperative, by removing the parts of the structure that generate local instability and hence promote non-cooperativity, making it impossible to correlate specific intermediate states with the proposed structures because those intermediates are no longer observed. Arguments analogous to those above also apply to the idea of lengthening the loops: longer loops will destabilize neighboring strands, again most likely changing the order in which intermediates occur. Complicating the measurement further, decreasing the stability of this already marginally-stable protein could reduce the unfolding force to levels that are difficult to measure reliably—the unfolding forces are already very low compared to other proteins described in the literature.

[Unpublished Data Redacted by Editorial Team Upon Authorial Request]

A final approach we took to obtain additional evidence supporting our model of the folding was to compare our measurements to computational simulations. This approach, too, has been successfully applied by others in previous work (see, for example, Best *et al.* (2001) *Biophys. J.* 81:2344–2356, Mickler *et al.* (2007) *PNAS* 104:20268–20273, Puchner *et al.* (2008) *PNAS* 105:13385–13390, Bauer *et al.* (2015) *PNAS* 112:10389–10394, Zheng & Glenn (2015) *J. Chem. Phys.* 142:035101, among others). We followed the method for simulating mechanical unfolding in SMFS measurements recently demonstrated by Habibi *et al.* (*PLoS Comput. Biol.* (2016) 12:e1005211), who applied their method to the double loop deletion mutant of SOD1 (loops IV and VII deleted). We first replicated the work done by Habibi *et al.* on the double loop deletion mutant, obtaining the same answers they found and thus verifying our implementation of the method. We next applied the method to simulate unfolding of the full monomer that we measured with the optical tweezers. (Incidentally, the simulations of the full monomer differed from the loop-deletion mutant simulations, further supporting the notion that changing the loops is not a fruitful approach for deciphering the nature of the intermediate states.) We found that the unfolding proceeded via numerous intermediate structures, as in experiments: an average of 3.1 intermediate states per simulated pull compared to 2.7 in the experiment. The structures of the intermediate states observed in the simulations agreed remarkably well with the structures that we had proposed based on the experiments: almost all the structures postulated from the FECs were observed in the simulations. Just as importantly, when we mapped all the transitions that occurred between the intermediates in the unfolding FECs (see Fig 3A in the revised manuscript)

and compared them to the transitions observed in the simulations (Fig 5B in the revised manuscript), we again found very good agreement.

Together, we believe that the experimental validation of the stable-core β -hairpin intermediate and the strong agreement between experiment and simulation provide good evidence supporting the model of the folding that we proposed. We have revised the manuscript in the Results and Discussion sections to describe these new results (including new figures presenting data from the peptide and the simulations).

We emphasize that although our model involves intermediates that match up with unfolding or refolding each strand individually, we don't actually see all the intermediates in any single FEC—in most cases, we only see 2 or 3 intermediates out of the full pool of eight. There are a number of possible reasons for such a subsampling of the full intermediate population: (i) intermediates might be missed because their lifetime is too short for reliable detection; (ii) intermediates might be missed because they involve 'internal' reorganizations that don't change the end-to-end distance (as an example, if strands 5–7 unfold before strands 1 and 8, we would not see the unfolding of strands 5 through 7 because the end-to-end distance would still be the same owing to the folded strands 1 and 8), thus showing up only when a structure at a terminus is disrupted; and (iii) numerous parallel sub-pathways may exist in which variable parts of the protein fold quasi-cooperatively. The net result is that a strict strand-by-strand mechanism overstates the observed non-cooperativity, and that is not really our model. Instead, what we are proposing is that the protein samples amongst a pool of intermediates that covers all the steps of the strand-by-strand process. The figure in the original manuscript illustrating our model was thus incomplete: it showed what we believe the intermediates are and how they are related structurally, but it didn't show the variable pathways between them observed in the experiment. We have therefore revised the manuscript to add a figure showing a map of all the transitions observed between all the different states and the relative frequencies with which they occur (Fig 3A in the revised manuscript).

Finally, we have systematized the analysis of the FECs that led us to the structural picture of the intermediates that we originally presented in Fig 4 (now Fig 3B). In the revised manuscript, we present a list (Table S1) of all possible intermediate structures that we might expect to observe (subject to some physically reasonable constraints reflecting the properties of both the measurement and the protein), which we use as a basis for a network analysis testing all possible combinations of states that might explain the observed patterns of intermediates in the FECs. What we find is that our original proposal offers the minimal explanation of the data, accounting for over 80% of all the data—additional pathways add no more than 1–2%, showing that our proposed pathways are by far the most likely.

After our extensive revisions, the analysis in the manuscript is now organized as follows:

- 1) We first identify the number of intermediates via the cumulative ΔL_c distribution.
- 2) Intermediates are assigned via their ΔL_c value to one of I_1 through I_8 , and we construct the transition map showing all pathways between folded and unfolded states that were observed in the FECs.
- 3) We next identify the most likely structure(s) for each intermediate by testing the sequences of ΔL_c values observed in FECs against the values predicted for all possible structures, in a network analysis that identifies the most likely pathways for folding.

4) Finally, we test the picture that results from this analysis using measurements of peptides designed to mimic intermediates and by comparing the experiments to simulations, in order to validate our identification of the intermediates.

5) *Fig. S1, the activity of labeled SOD1: the authors need to provide more information about the samples. How are the labeled SOD1 prepared and purified? What is their purity?*

We only referenced the method for labeling SOD1 in the original manuscript, rather than describing it in detail, because it has been published previously. As requested, however, we have revised the Methods section to describe the labelling of the SOD1 in greater detail. Briefly, purified SOD1 was reduced with TCEP, and then the reduced thiols were activated with DTDP after removing excess TCEP by spin filtration. The activated protein was then purified from excess DTDP by spin filtration and reacted with sulfhydryl-labeled DNA handles prepared by PCR. The labelling effectiveness was tested by mass spectrometry. Peaks were observed in the mass spectrum only at 19.633 kDa (corresponding to unlabelled SOD1) and 19.853 kDa (corresponding to SOD1 labelled with 2 DTDPs). The relative peak magnitudes indicated that about 30% of the molecules were labelled under the conditions we used.

Reviewer #2:

Misfolding of Cu/Zn-superoxide dismutase (SOD1) triggered by amino acid substitutions has been proposed as a major pathological process in amyotrophic lateral sclerosis. Many researchers have attempted so far to understand the folding process of SOD1 proteins with various experimental/theoretical methods. In this paper, the authors have characterized the folding and misfolding of SOD1 proteins with single-molecule force spectroscopy. They artificially extended a single SOD1 molecule by using dual-trap tweezers, and the force-extension curves of SOD1 unfolding/refolding were analyzed. The curves were found to be interrupted by “rips”, based upon which the authors proposed the folding intermediates of SOD1.

The referee introduces here the notion—repeated in many of the comments below—that our study is somehow “artificial” because of the way our measurements were done. Given that this idea of artificiality apparently underlies many of the referee’s objections, we feel that it is important to address it up front, separately from the individual comments below.

We note that **every** *in vitro* assay of protein folding and structure is “artificial”, in the sense that they all study the protein outside of its normal cellular environment. Applying forces to the termini of a protein is no more “artificial” than subjecting the protein to chemical denaturants, imposing large temperature or pressure jumps on it, or crystallizing it to determine its structure. Indeed, most of the standard assays that form the backbone of our knowledge of protein structure and folding are, if anything, *more* artificial than force-based assays like the one we use: proteins in the cell are never exposed to high levels of chemical denaturant (as in urea- or guanadine-induced unfolding and refolding), nor do they undergo large and rapid temperature or pressure jumps, nor are they crystallized, but they do experience forces on the scale of a few pN, for example from collisions, shear forces and other hydrodynamic effects, membrane translocation, ligand binding, the action of molecular motors, etc... (see, among others, Javadi *et al* (2013) *Physiology* 28:9–17, Jagannathan & Marqusee (2013) *Biopolymers* 99:860–869). Nevertheless, *in vitro* assays—despite involving measurements done under conditions that differ from what is

found in live cells—can still reveal information about the properties of the protein that is valuable to understanding how it behaves. Furthermore, the controlled conditions of *in vitro* assays can enable the discovery of information that is simply impossible to obtain from a protein in the complex cellular environment, which is why *in vitro* assays continue to form the backbone of protein science. Criticizing a particular *in vitro* assay as being “artificial” is thus not justified, unless one is willing to reject the entire corpus of *in vitro* work (the vast majority of protein science) for the same reason.

The paper appears to be interesting but cannot be recommended for publication because of the following reasons.

1: The results in this paper do not describe (or actually, do not have any relation with) the prion-like activity of misfolded SOD1.

We respectfully point out that we do not claim in the manuscript that the point of our work is to describe the prion-like activity of SOD1. The focus of the work is on examining the folding and unfolding of SOD1 monomers at the single-molecule level, which allows higher resolution and sensitivity than previously possible, in order to map out folding pathways and misfolded states. This focus is described quite clearly and explicitly in the title (“Partially native intermediates mediate misfolding of SOD1”) and in the introduction section (“Here we examine SOD1 folding using single-molecule force spectroscopy, [...] aiming to map the pathways for both native folding and misfolding, and determine how they are related”). Of course, a lot of the interest in SOD1 is ultimately driven by its prion-like abilities, but before those abilities can be understood, it is essential to have as complete an understanding of SOD1’s basic folding properties, the elementary underpinnings of the misfolding, and how misfolding relates to native folding, which are precisely the issues that we have addressed here.

We do state that our results have suggestive implications for understanding the basis of prion-like conversion. Our remarks here are supported by perfectly sound reasoning: the fact that the misfolding-prone parts of the native structure are the least stable and can unfold non-cooperatively provides an obvious possible route for converting the native fold to a misfold, and such a route would presumably be easier than if the entire native structure first had to be unfolded cooperatively before conversion could take place. These concluding remarks, however, are not the primary message of the paper, rather they are forward-looking statements pointing out implications of the work that could be followed up in future research. We feel that it is inappropriate to judge a paper as unsuitable for publication because it does not address questions that are not, in fact, the goal of the work.

If, instead, the referee is implying that only manuscripts that address the prion-like activity of SOD1 are publishable, we respectfully point out that this notion is contradicted by the considerable number of papers that have been published on SOD1 folding and misfolding *without* addressing prion-like activity, including in high-impact journals.

2: While the authors described that single-molecule methods have not been applied to study misfolding in SOD1, Solsona et al. already examined SOD1 folding process with single-molecule force clamp spectroscopy (J Biol Chem 2014 289 26722).

Our statement that misfolding of SOD1 has not been studied in the single-molecule limit is, we believe, strictly correct: the paper mentioned by the referee did not, in fact, investigate SOD1

misfolding. The manuscript by Solsona *et al*, entitled “Altered thiol chemistry in human amyotrophic lateral sclerosis-linked mutants of superoxide dismutase 1”, studied disulfide isomerization reactions in mechanically unfolded SOD1. Solsona *et al* focused on the mechanical cleavage of disulfide bonds. They observed unfolding of the protein structure (as opposed to cleavage of disulfide bonds) in only a small fraction of the molecules they measured (less than 10%), and they did not identify any examples where they found that SOD1 was misfolded. They did show that the native disulfide bond can isomerize with C111 when the rest of the protein is unfolded, which might in principle lead to a misfolded structure upon refolding. However, they did not demonstrate any examples where the protein was refolded in such a way. Our statement is thus well-justified by the factual content of the paper by Solsona *et al*.

Nevertheless, we have revised our statement to mention the previous single-molecule work on disulfide-bond isomerization in SOD1.

3: The “tweezers” are attached to the N- and C-termini of SOD1; therefore, in this experiment, the site from which the unfolding starts is limited to the N- and C-termini. It is possible that SOD1 starts to unfold at the site(s) other than the N- and C-termini, but such unfolding process cannot be evaluated by the method in this study.

We agree with the referee that where the linkers are attached to the protein can impose certain constraints on what can be observed, and structural changes that leave the end-to-end distance unchanged are largely invisible in the experiment (although their effects may in some cases be detectable via the effects on the lifetimes of the states). However, such constraints still leave open the detection of numerous possible ways for the protein to unfold. In particular, the unfolding is *not* limited to starting at the N or C terminus, as the referee asserts. If the least stable part of the structure were some interface between two domains or sub-domains away from the termini, for example, then the structure could rupture at that interface, creating two intermediate pieces of the original structure that would then each unfold independently. Such behavior has been seen previously in SMFS experiments (see, for example, Jahn *et al.* (2016) *PNAS* 113:1232–1237). In the case of SOD1, one could imagine as a possibility that the sheet formed by strands 1–3 might dissociate from the rest of the structure as a first step. Such a situation would be clearly detected in our measurement: it would lead to $\Delta L_c \sim 6.5$ nm, inconsistent with our data.

As a consequence, the fact that what we see is consistent with sequential peeling of the strands is, indeed, meaningful—it did not have to happen this way, contrary to the referee’s assertion. As we mentioned above, it is likely true that we are not detecting all the intermediates that are present in SOD1 folding (as also discussed in response to point 1b from referee 3), because of the limitations of the measurement. Nevertheless, even if we do not detect every intermediate, our measurements still reveal far more detail about the intermediates in the folding than could be detected in previous ensemble measurements (on the basis of which SOD1 has been thought to be a two-state folder), and there is thus a lot to be learned from them about how SOD1 monomers behave.

Finally, we feel that it is important to point out an important subtext to these issues: *every* experimental method imposes certain constraints on what can be observed, from both the choice of properties that are measured (whether length, FRET efficiency, Trp fluorescence, or something else) and the conditions under which the measurements are done (whether under the influence of a local perturbation like force, a global one like urea, or something mixed like

pressure). Different methods are complementary, revealing different facets of the full behavior of the protein. If two methods yield different results, it does not mean that one of the results is “right” and the other is “wrong”; rather, each is correct under the conditions applicable for that method. Moreover, even if different methods yield results that differ on the surface, often one can find deeper areas where the results do indeed overlap. Such is the case in our work, where—despite the differences between our single-molecule force methods and the methods used in previous ensemble studies—there are numerous points of agreement in terms of the architecture of the folding.

In order to clarify the analysis of the intermediates and show that we have indeed included the possibility of intermediates that do not purely involve unfolding from the ends, we have added a network analysis of the intermediates observed in the FECs, comparing the observed sequence of length changes to the values that could be expected from all realizable combinations of folded and unfolded strands in SOD1 (including possibilities in which some of the middle strands unfold early on, leaving disjunct folded regions separated by an unfolded region). This analysis finds that substructures involving disjunct folded regions—where the unfolding occurred at a location other than the C- or N-terminal edges of the structure—are inconsistent with the observations. Indeed, the structures we originally identified for the intermediates are found to be the most likely explanation, as they are the smallest set of structures needed to account for the vast majority of the data. This new analysis is described in the Results and Methods sections of the revised manuscript. We have also added an explicit statement in the Discussion section that our measurement is limited to detecting structures in which the end-to-end length changes.

Also similarly, the N- and C-termini of SOD1 would interact with each other during the early process of SOD1 refolding. The experimental system in this study is therefore supposed to be quite artificial.

First, we note that the referee’s assertion that interactions between the N and C termini are critical to initiating the folding of SOD1 is something that can be tested directly with the force spectroscopy data: if this assertion were true, then we would be able to detect it in the FECs because it would result in an absence of intermediate states during refolding, coupled effectively with an energy barrier (transition state) that is very close to the native state. Exactly this result has been observed previously in force spectroscopy measurements of numerous different proteins—including those in which both the C and N termini consist of β -strands that are adjacent to each other (as in SOD1), such as the SH3 domain studied by Susan Marqusee’s group (Guinn *et al.* (2015) *Nat. Commun.* 6:6861), the protein GB1 studied by Hongbin Li’s lab (Cao & Li (2007) *Nat. Mater.* 6:109–114), scaffoldin (Valbuena *et al.* (2009) *PNAS* 106:13791–13796), and ubiquitin (Fernandez & Li (2004) *Science* 303:1674–1678). There is thus no intrinsic impediment to observing such behavior in force spectroscopy measurements. The fact that we do not see such behavior with SOD1 is therefore significant: it indicates that interactions between the N and C termini are not, in fact, critical in the early stages of refolding of SOD1, contrary to the referee’s claim. Furthermore, the referee’s assertion is not supported by previous ensemble biochemical studies or computational simulations. According to both phi-analysis (Nordlund & Oliveberg (2006) *PNAS* 103:10218–10223) and simulations (Ding & Dokholyan (2008) *PNAS* 105:19696–19701), the C terminus in particular seems to be largely unstructured in the transition state, which is inconsistent with the referee’s statement that interactions between the two termini are critical early in refolding.

The referee's claim that our measurement is "quite artificial" because it prevents the termini from interacting during the early stages of refolding, as the critical step in refolding, is thus not supported by the evidence in the literature: there are multiple publications demonstrating that force spectroscopy measurements can detect transition states that involve interactions between the termini even when pulling on the proteins from their termini, and studies using other methods also point to a lack of interactions between the termini in the transition state for SOD1 folding.

The authors should examine several permutants of SOD1.

The referee does not specify the intended purpose for this suggestion, but we assume that the goal is to help identify the order in which the structural components of SOD1 unfold and refold. Unfortunately, examining circular permutants of SOD1 would almost certainly be unhelpful for accomplishing this goal, because it has previously been shown that the chain topology helps to control the folding cooperativity. Shank *et al.* (*Nature* (2010) 465:637–640) showed directly that circular permutation can change the coupling between different parts of the protein and thereby change the number of intermediate states observed in the measurements. In the case of the unfolding and refolding of apo-SOD1, where the interactions between structural components are not particularly strong (since the protein is not very stable), small changes in the interactions induced by changes in the chain topology from the circular permutation would almost certainly lead to large changes in the outcome of the folding. For example, we could expect the cooperativity (number of intermediates) to change, as in the work of Shank *et al.*, or we could expect the ordering of the intermediate states to change (because of changes in the strength of the interactions between different parts of the structure). Without knowing in advance the precise nature of these changes—which definitely are *not* known—it would be impossible to interpret the results of measurements on permutants and draw reliable conclusions.

Although there are no doubt interesting questions about protein folding that can be explored through measurements of circular permutants, permutants will not be helpful for validating our model of the folding of SOD1. The strategies we outlined above in response to point 4 of referee 1 provide a more effective and reliable basis for doing so, hence we have focused our efforts on these approaches.

However, I think that the pathologically relevant misfolded conformation(s) cannot be revealed by the experimental methods in this study.

We respectfully note that this assertion is neither supported by any evidence, nor is it particularly relevant to the evaluation of our work. Firstly, given that this is the first manuscript describing a force spectroscopy study of misfolding in SOD1, it seems premature to render a verdict already on whether the method is capable of characterizing pathologically relevant misfolded states. Secondly, we reiterate that our goal in this work is not to identify pathogenic misfolded states, but rather to map out the folding behavior of SOD1 monomers and how native folding is related to the possible non-native states that SOD1 can form, hence it is inappropriate to judge the method for its ability to do something that is not even under study. Most importantly, however, we note that the pathogenesis of ALS (and the role played by SOD1) is still very much in dispute in the literature (see, for example, Andersen & Al-Chalabi (2011) *Nat. Rev. Neurology* 7: 603–615), and so far as we know *no experimental method has yet identified a misfolded conformer of SOD1 that is proven to be pathogenic!* Even setting aside the fact that we are not claiming to detect any pathogenically relevant structures, it seems inappropriate to pre-judge as a failure a

method that is among the few that have not yet been applied to the problem, when the other methods that have been used so far have all been unsuccessful.

4: The authors described that SOD1 folding/refolding are characterized by the high degree of non-cooperativity; however, the high degree of non-cooperativity appears to be artificial due to the experimental method, single-molecule force clamp spectroscopy.

The referee's assertion that we observe non-cooperative behavior because our approach is "artificial" (and by implication, that the cooperative transitions observed in most ensemble studies of SOD1 are somehow "natural") is not justified, as we have argued above. In every experimental approach to folding, one perturbs the protein out of its most stable conformation and then examines its conformational dynamics as the protein reacts to the perturbation. The perturbation induced by locally applied forces is not any more "artificial" than the perturbation induced by other means, such as urea denaturation or temperature jumps (indeed, it is arguably less so, as discussed above). It is just less commonly used in the literature, and hence presumably less familiar (perhaps leading to the unwarranted perception that it is therefore "artificial"). Furthermore, if the referee's assertion that force spectroscopy artificially induces non-cooperativity in folding were true, one would expect it to do so ubiquitously, producing "artificial" intermediates in most if not all proteins studied. Yet that is manifestly not the case: many proteins studied by force spectroscopy show no evidence of intermediates—for example, apomyoglobin (Elms *et al.* (2012) *PNAS* 109:3796–3801), prion protein (Yu *et al.* (2012) *PNAS* 109:5283–5288), the SH3 domain (Jagannathan *et al.* (2012) *PNAS* 109:17820–17825), ubiquitin (Fernandez & Li *Science* 2004), GB1 (Cao & Li *Nat. Mater.* 2007), cold-shock protein (Hoffmann *et al.* (2013) *J. Phys. Chem. B* 117:1819–1826), scaffoldin (Valbuena *et al.* *PNAS* 2009), and many others. The referee's claim is thus not supported by the evidence in the literature.

We note that using different experimental methods is valuable precisely because they can be sensitive to different aspects of the behavior. Single-molecule methods, for example, are generally viewed as being more sensitive to the presence of intermediates than ensemble methods (see, for example, Cecconi *et al.* (2005) *Science* 117:1819–1826, Neuman *et al.* (2008) *Nat. Methods* 5:491–505), so it is not surprising that we detect intermediates that were not observed in previous ensemble work. We very purposefully discussed how the single-molecule results compare to previous ensemble studies in considerable detail in order to address this issue directly—different experimental methods give different views of the folding process, but commonalities can still be discerned.

We have revised the Discussion section to clarify that non-cooperativity is not a general feature of SMFS measurements.

5: The preparation of SOD1 samples was not clearly described.

For the construct design and protein purification, we provided in the original manuscript the information that is typically included in publications: the mutations that were made, the plasmid used for expression, the bacterial growth conditions, how cells were treated, and the conditions for the affinity-column purification (including detailed buffer conditions). The functionalization of the protein to create DNA-SOD1 chimeras was only described briefly, however. We have revised the Methods section to provide a more complete description of the functionalization procedure and the controls done to ensure proper functionalization.

Both metallation and thiol-disulfide status of SOD1 (or actually, artificially monomerized SOD1) were not experimentally confirmed.

With respect, these assertions are incorrect. We know that the native disulfide must be reduced in the measurements because of the total contour length change that we observe. The forces we applied (up to ~15 pN) are more than an order of magnitude too small to break disulfide bonds (as seen in the work by Fernandez and colleagues, where disulfide bonds were difficult to break at ~200 pN unless DTT was added, see Wiita *et al.* (2006) *PNAS* 103:7222–7227). Hence if the disulfide bond had been present in the molecules we measured, we would only have been able to stretch out the unfolded protein up to the disulfide bond. Since this bond is between residues 57 and 146, the 90-amino acid segment between these residues would not get stretched out under the tension, and we would therefore expect a total contour length change of only 22.5 nm (as illustrated in Fig R4). Given that we see a total length change of 54 nm, the disulfide bond must, in fact, be absent.

Figure R4: Effect of disulfide bond on FECs. If the bond is reduced, then the entire polypeptide chain can be stretched out upon unfolding, leading to the observed ΔL_c . If the bond is present, then only a portion of the polypeptide chain could be stretched upon unfolding, reducing ΔL_c by more than half.

With regards to the metallation status of the protein, we know that it must be de-metallated for two reasons:

- (1) First, ensemble studies have shown that metallation hugely increases the stability of SOD1, to about 30 kcal/mol (Stathopoulos *et al.* (2006) *J. Biol. Chem.* 281:6184–6193). The low unfolding forces we observe—lower than for any other β -structured protein that has been reported, as far as we are aware (see, for example, Hoffmann & Dougan (2012) *Chem Soc Rev* 41:4781–4796)—is inconsistent with such a high stability. For comparison, GFP, another stable β -barrel protein (stability of ~15–20 kcal/mol), unfolds at a force of ~100 pN or more (Dietz & Rief (2004), *PNAS* 101:16192–16197). Although Solsona *et al.* (2014) only present qualitative results for unfolding holo-SOD1, they do report observing unfolding of the protein in some cases at forces of over 100 pN, implying that the unfolding force for holo-SOD1 is indeed much higher than the <10 pN values we observe.
- (2) Second, the presence of EDTA as a chelating agent in the buffer during the SMFS experiments also ensures that it is not possible for the SOD1 to remain metallated. Even if natively folded SOD1 molecules retained ions bound to the protein after the sample-preparation stage, these ions would unbind once the protein was unfolded completely in the first pull, since unfolding destroys the binding sites. The presence of EDTA in the solution would then ensure that any ions left over after purification could never rebind to the protein after the first pull. Notably, we see no difference between the unfolding forces in the first pull and those in all subsequent pulls, indicating that the protein must be de-metallated throughout our SMFS measurements.

We have revised the manuscript in the Results and Methods sections to explain how we know that the protein is in the reduced apo state in our measurements.

Also, SOD1 in this study was reduced with TCEP, so the intramolecular disulfide bond was supposed to be reduced. It is quite obscure where in SOD1 the sulfhydryl-labeled DNA handles were attached.

As explained in the Methods section, the DNA handles were attached to the Cys residues added to each terminus of the protein. To make this point clearer, we have added labels to Fig 1A indicating the N- and C-terminal Cys attachment points. We know that only the terminal cysteines are labelled because of controls measuring the labelling stoichiometry, controls testing the location of the labelling, and the fact that only handles attached to the two terminal cysteines would produce a total length change equal to full contour length of the protein (see response to point 2b of referee 3 for more details).

Reviewer #3:

This manuscript by Mojumdar et al. uses high-resolution single molecule optical trap spectroscopy to monitor the unfolding and refolding of demetalated, monomeric SOD. Unfolding and refolding are monitored by force ramp experiments and examination/analysis of the force extension curves. The authors find a very non-cooperative multi-step unfolding and refolding curve. Using Occam's razor to interpret these 'rips' or changes in extension, the authors propose a stepwise mechanism of unfolding and refolding for this protein. They also find evidence for misfolding from some of these intermediates, which together with their structural interpretation suggest regions to target in the misfolding trajectory of this medically important protein.

Overall, I like this manuscript and think it is of interest to a broad audience. I am somewhat concerned that it does not present very much data (just FECs on the one protein) and might be more impactful if it made some mutations to test the structural pathway it proposes. I assume those experiments will be forthcoming and given the general interest in the unfolding of SOD, do think it represents an important advancement.

Referee 1 raised similar concerns in his/her point 4. We provided a full response above, but briefly: we have added data from a truncation mutant that consists of one of the β -hairpins that we posit form part of the 'stable core', showing that it does indeed fold independently with the length observed in SOD1. Furthermore, we simulated the unfolding computationally and found excellent agreement with the proposed model of the folding pathway shown in Fig 4.

I do have a few question/concerns that should be addressed.

1. Data analysis/interpretation:

a. How were the rips identified in the force ramp experiments? Was this done manually or by a nonbiased algorithm?

The rips were identified manually, because of the technical difficulties involved with analyzing data with a large but variable number of intermediates. That said, some quantitative criteria were part of the identification procedure: the intermediate had to persist for a minimum of 10–15 ms

to be counted as present, and the extension and force changes during the rip had to be larger than the noise fluctuations in the parts of the FECs without rips and in measurements of DNA handles only. We are currently developing a more automated algorithmic approach based on clustering, but it is still being refined to reduce the error rate.

We have revised the Methods section to discuss how the rips were identified. We have also added a new figure (Fig S4) showing measurements of the DNA handles only, for comparison to the measurements with the protein present.

The representative curves seem to show some noise on the level of the size of the rips – are those very transient ‘equilibrium’ visitations to the intermediate?

The referee is correct: quasi-equilibrium hopping between states is observed fairly frequently in the FECs, indicating that the curves are not very far from equilibrium. The observations of quasi-equilibrium hopping are also consistent with the fact that the refolding forces are very close to the unfolding forces, another indication that the measurements are done in near-equilibrium conditions. We have revised the Results section to mention these points explicitly.

b. The author’s explanation of their data is based on a very native ‘GO-like’ model – ie the unfolding of individual strands off the edges of the beta sheet. But isn’t this the only kind of intermediate one could see with this method (regions coming off from the edges)? If an intermediate formed by unfolding an internal beta strand or a hairpin, there would be no change in extension – so how would this be observed? Is it possible that there are many more intermediates not seen in these FEC curves?

The referee is correct that only structural changes that alter the experimental observable—in this case, the end-to-end extension—can be detected directly in our measurements. (Although one must recognize that such a limitation, of course, applies to all experiments, regardless of the specific method used!) Hence, for example, if strands 4 and 5 were to unfold first while the rest of the protein remained folded, then we would not detect this intermediate because it does not change the distance between the folded termini. The referee is thus correct that the number of intermediates we see is just a lower bound: more could indeed be present. However, we note that an ‘internal unfolding’ event like this *would* show up in our data, once the outer parts of the protein had unfolded: at this point, the ΔL_c associated with the already-internally-unfolded portion of the structure would produce an extra-large rupture, bypassing the intermediates that would otherwise have been associated with peeling off strands 4 and 5 individually. Such internal unfolding could help explain, in part, why the intermediates are ‘subsamped’ on any given FEC (*i.e.*, we don’t see each intermediate on every curve), but subsampling could also arise from ‘missed events’ (where an intermediate appears to be skipped owing to time-resolution limits) as well as from partial cooperativity (where 2 or 3 strands might unfold or refold together, for example, but not always in the same combination).

Even given this limitation that only structural changes that alter the extension can be detected, however, it does not follow that the only intermediates one can observe are those in which the edges of the structure are peeled away, as the referee suggests. For example, imagine a structure consisting of two sheets stacked on top of each other, connected by a loop. Unstacking of the sheets, without unfolding any of the strands within a sheet, would generate a very detectable extension change as the sheets are pulled apart. Thus peeling from the edge of a sheet is not the only possible transition that needs to be taken into account when explaining force spectroscopy

data. An example one could imagine occurring in SOD1 would be an unfolding transition from the native state in which strands 4–6 unfold leaving the sheet β 1–3 and hairpin β 7–8 as independent, dissociated ‘substructures’.

We emphasize that the model of the folding we proposed in the discussion section, based on sequential unfolding/refolding of strands in the native structure, is just the simplest explanation of all the observed data (as well as being consistent with the new measurements of the strand 2–3 hairpin and the simulations). To strengthen the reasoning that led us to this model, we have added a systematic network analysis of the patterns of length changes in the FECs that compares them to the lengths for all possible combinations of folded and unfolded strands in SOD1, in order to identify which sequences of structures are consistent with the data. Our model is the most likely explanation of the data. In particular, although some of the lengths that we observe for individual intermediates are consistent with substructures in which SOD1 has ‘fractured’ in the middle to form disjunct folded regions, the patterns of length changes we see are not consistent with the subsequent independent unfolding of these fragments (as illustrated in the new Fig S1). More complex explanations can of course always be postulated, but normal scientific practice is to use the most parsimonious explanation of the data, which is what motivated the model we presented.

We have revised the manuscript to explain these issues more clearly.

c. I am confused by the fact that the same intermediates are seen in both folding and refolding? For example, in Figure 1 c,d and e the last unfolding rips is just under 10pN and the first refolding rip is also around that force. But these are not equilibrium experiments – shouldn't refolding occur at a lower force than unfolding? Perhaps this is just a result of showing representative curves? Shouldn't there need to be a barrier separating the two in order to transiently form an intermediate – I would assume that you would not see all the same barriers in each direction. Since the authors are proposing a very ordered optional set of steps, perhaps they could put them on some energy (or even force) diagram to help me (and others) understand.

As described above in response to point 1a, the experiments are actually in near-equilibrium, as seen empirically from the not-infrequent examples of “hopping” back and forth between nearby states in the FECs as well as from the similarity between the unfolding and refolding forces. The fact that the same intermediates are seen in unfolding and refolding is thus just another consequence of the measurements being in near-equilibrium. How far a measurement is from equilibrium depends on a balance between the rate of change of the force (the loading rate) and the folding/unfolding rate (or equivalently, how long it takes the force to change compared to the lifetime of the state being occupied). Here, the lifetimes of the intermediates are all fairly short: on the order of 0.01–0.1 s, during which time the force only changes by a fairly small amount (about 0.2–2 pN). As a result, there is not much hysteresis between the unfolding and refolding curves. Note that the short lifetimes imply that the barriers between intermediates are fairly small. We have revised the manuscript in the Results and Discussion sections to point out explicitly the significance of the examples of hopping in the FECs—that the transitions are in quasi-equilibrium, implying small barriers.

Regarding the “very ordered” set of steps in our model of the folding, we regret that our figure illustrating the model provided an incomplete picture of what we are proposing. The figure showed our model of how all the intermediates are related structurally (*i.e.* what structural changes are required to go from one intermediate to another). However, it did not show the

varying arrangements of these intermediates within the individual FECs, where a variable subset of all possible intermediates were seen in any given curve. To make clear how the protein navigated through all the different intermediates, we made a transition map indicating all of the pairwise transitions between different states in unfolding and refolding FECs along with their frequency of occurrence (Fig 3A in the revised manuscript). This map gives a clearer picture of all of the alternative pathways through the intermediate states that are reflected in the data. Note that this map is empirical—it makes no assumptions about the nature or structure of the intermediate states, simply assigning the state based on the value of the unfolded contour length in that intermediate.

Finally, regarding the referee's suggestion to provide an energy diagram, we had originally hoped to be able to perform a quantitative energy landscape reconstruction using one of the many approaches available in the literature (see, for example, Woodside & Block (2014) *Annu Rev Biophys* 43:19–39). However, for technical reasons none of these methods can be applied reliably. One problem is that there are too many states too close together in length and they fluctuate too rapidly (reconstruction methods work best when states are well-separated in length and have reasonably large barriers between them). A second problem is the fact that SOD1 samples different sets of intermediate states in different pulls, which implies that there may be multiple pathways through the landscape and hence a one-dimensional treatment of the landscape will not be sufficient. To date, there are few analytical methods capable of dealing with multiple pathways and dimensions—work by Dudko and colleagues comes to mind (Suzuki & Dudko (2010) *Phys Rev Lett* 104:048101, Pierser & Dudko (2017) *Phys Rev Lett* 118:088101)—and none of this work extends reliably to such a profusion of states and pathways as we observe in our measurements. Additional work will therefore be needed to reconstruct the landscape for SOD1.

Although we are unable to provide a landscape reconstruction, we believe that the transition maps for unfolding and refolding that we have added to the manuscript (Fig 3A in the revised manuscript) should achieve the goal sought by the referee—providing a clearer understanding of the set of steps taken during the unfolding and refolding.

d. I think the authors could explain more clearly how they know the misfolds are distinct from the intermediate states. The contour lengths presented in table 2 for M1, M2 and M3 are similar to the contour length of these intermediates. I'm sure the authors have thought about this, but more discussion in the paper of how they know the misfolded states are distinct from the intermediate states would be useful.

The misfolded states can be distinguished from the on-pathway intermediates in two ways: via different contour lengths, and via different kinetics (lifetimes). The most obvious difference is in the lifetimes: whereas the on-pathway intermediates have relatively short lifetimes of only about 0.01–0.1 s in the FECs, the misfolded states have lifetimes of many seconds. As discussed in response to point 2 of referee 1, we have now measured the average lifetime of the misfolded states to be ~ 6 s, over 100 times longer than the typical on-pathway intermediate lifetime. With respect to the contour lengths, the two most common misfolded states (M_1 and M_3) are distinct from any of the on-pathway intermediates; the rarest misfolded state (M_2) has a length similar to that for one of the on-pathway intermediates, but this state can still be distinguished via its long lifetime. To illustrate concretely how the lengths of the misfolded states relate to the lengths of the intermediates, we plotted the contour lengths associated with misfolding on the same graph

as the cumulative ΔL_c for all the on-pathway intermediates (see Fig S2 in the revised manuscript). We have also revised the figure showing the contour lengths associated with the misfolded states (Fig 6C), to present the results in terms of the contour length difference as measured from the native state (*i.e.* the cumulative unfolded contour length), so that these lengths can be compared more easily to the lengths of the on-pathway intermediate states.

We have revised the Results section in the manuscript to discuss how the misfolded states are distinguished from on-pathway intermediates.

e. On page 8-9, they compare their results to ensemble studies. Please clarify if these ensemble studies are also on an un-metalated monomer.

Yes, these ensemble studies were done on apo-SOD1 monomers. We have clarified this point in the revised manuscript.

2. Sample prep:

a. How do they know that the protein is effectively demetalated? Is it already known that simple addition of chelators is enough? Would the authors get the same result by first unfolding to high force (and in essence removing the bound metals) and then recording future ramps (or is the metalated form too stable to unfold at the accessible forces?).

As explained in response to point 5 of referee 2, there are two reasons we can be assured that the protein is de-metallated: the low unfolding forces observed (holo-SOD1 is very much more stable than apo, and should unfold at much higher force), and the fact that the chelator ensures that any residual metal ions that might have remained bound to a given protein molecule will be sequestered after that molecule is first unfolded.

b. The ability to attach DNA handles to a protein with several 'internal' cysteines seems novel and important. The authors claim to mutate the surface exposed cysteines and, by using mass spec, they can determine that the handles are only interacting with the engineered terminal cysteines. I am not exactly sure how they did this (I imagine, they showed dttp addition (by mass spec) on constructs without the engineered cysteines and no DTDP on those without the engineered cysteines). This should be explained more clearly and perhaps added to supplementary material.

We have actually applied this method to another protein containing internal Cys residues before, the prion protein PrP (see Gupta *et al.* (2016) *Nat. Commun.* 7:12058). The referee is correct, the approach taken is to quantify the addition of DTDP to the protein by mass spectrometry. With the 2 terminal cysteines, 2 DTDP molecules react with the protein; without the terminal cysteines present, no DTDP react with the protein. A final verification that the handles are attached to the terminal cysteines (and not the internal ones) comes from the length change upon unfolding: the length change matches the distance expected for the full native structure, instead of the shorter distance that would be observed if the handles attached to an internal cysteine. In our work on SOD1, we first tested the labelling of the wild-type protein (with solvent-exposed Cys residues removed) exactly as above, measuring the stoichiometry of DTDP addition to the versions with and without added terminal cysteines via mass spectrometry. We found that exactly 2 (not 1, 3, or 4) DTDP reacted with the version with terminal cysteines, but DTDP did not react with the version without terminal cysteines. We then measured the DTDP addition stoichiometry to the

monomeric mutant with terminal cysteines, finding again that 2 (not 1, 3, or 4) DTDP reacted with the protein. We did not repeat the control with no terminal cysteines for the monomeric mutant, relying instead on the length measurements to verify that handles were indeed bound to the terminal cysteines. We have revised the Methods section to explain how we verified that only the terminal Cys residues were labelled.

c. I suggest adding information about the protein – that is human SOD with some mutations to remove external cysteines (defined by what? I assume solvent exposed surface area) and prevent dimerization. This information can be found in the methods – but it is something a casual reader would want to know.

We have revised the manuscript as suggested, at the beginning of the Results section. We note that the ‘pseudo-wild-type’ form of the protein with the solvent-exposed Cys residues removed is a version that is quite standard in the literature and has been heavily studied by ensemble biochemical and biophysical assays (see for example Lindberg *et al PNAS* 2004).

List of most significant changes to the manuscript:

1. New analysis of the FECs to create a map showing all transitions between states was added, in response to referees 1 and 3.
2. New analysis systematizing the identification of consensus pathways and possible structures for the observed intermediates was added, in response to all referees.
3. New measurements of a peptide were added to validate the proposed structure of an intermediate state, in response to referees 1 and 3.
4. Computational simulations of the unfolding under applied force were added as another test of the proposed structures for the intermediate states, in response to referees 1 and 3.
5. Measurements of the lifetime of the misfolded state were added, in response to referee 1.
6. A new author (Z. Scholl) was added, to reflect the significant contribution he made to the revised manuscript.
7. A reference to a previous single-molecule study of disulfide bond isomerization in SOD1 was added to the Introduction, in response to referee 2.
8. The evidence supporting the fact that SOD1 is in the apo reduced state in the measurements was added to the Results and Methods sections, in response to referees 2 and 3.
9. The number of molecules studied was listed and the similarity between molecules was described (Results, Methods, SI), in response to referee 1.
10. The quasi-equilibrium nature of the measurements was noted (Results) and its implications discussed (Discussion section), in response to referee 3.
11. The motivation for analyzing the data in terms of the cumulative unfolded contour length was explained (Results, Methods), in response to referee 1.
12. The reasoning leading to the proposed structures for the intermediates was explained in greater detail (Results, Methods, SI), in response to referees 1 and 3.
13. The nature of the misfolded states and how they compared to on-pathway intermediates was outlined (Results section), in response to referees 1 and 3.
14. The fact that SMFS does not habitually induce non-cooperative behavior was noted (Discussion section), in response to referee 2.
15. The model of the folding was clarified (Discussion), in response to referees 1 and 3.
16. The protein labelling method and tests of labelling outcomes were described more fully (Methods), in response to referees 2 and 3.
17. The FEC fitting process was described in more detail (Methods), in response to referee 3.
18. New references were added to support the new material. Some old references were removed to provide space for the new references.
19. The reporting of the misfolded state lengths in Table 2 was change, in response to referee 3.
20. Fig 1A was modified to show the location of the terminal Cys residues used for handle attachment, in response to referee 2.
21. Fig 3 was changed to show the new transition maps for native unfolding and refolding (3A), and the proposed structures for intermediate states (formerly Fig 4). A new Fig. 4 was added to show FECs from the peptide. A new Fig 5 was added to show the results from the computational simulations. The misfolding results were moved to Fig 6, and a new panel was added to show the lifetime data.
22. Supporting information detailing the simulations and pathway analysis was added.
23. New SI figures were added, illustrating the pathway analysis (S1), comparing the contour lengths of the misfolded states to those of the on-pathway intermediates (S2), and showing FECs of DNA handles only (S4), in response to reviewers 1 and 3.

Reviewers' comments:

Reviewer #1 (Remarks to the Author):

In the revised manuscript, Mojumdar et al carried out additional experiments and attempted to address the questions I raised in the first round of review. However, I remain unconvinced by the response and additional experimental evidence provided by the authors.

1) With regard to the deltaLc analysis: although the cumulative deltaLc analysis could be potentially advantageous in certain cases, a fundamental assumption of this method is that protein unfolding follows a particular sequence of events. This assumption is unfounded. An example in this regard can be found in Stigler et al Science, 2011, 334, 512. For a protein that is prone to misfolding, it is hard to imagine that misfolding will follow a particular well-defined pathway. Without proving the validity of this assumption, my concern remains: "Reporting cumulative deltaLc may lead to wrong grouping of the events. I suspect that the wide deltaLc distribution for some intermediate states is simply due to the grouping of different and unrelated intermediate states into one population."

2) "The data were measured from 6 different molecules, with an average of about 400 pulls each. This is enough data from each molecule to check that the histograms from different molecules are indeed similar"

I am concerned about the extremely low number of molecules reported in this work. Although each molecule contains ~400 pulls, this fact does not prove that these six molecules reflect the true behaviors of SOD1. More evidence/data is needed from different molecules.

3) Regarding experiments attempted to provide more experimental evidence for the peeling model:

I acknowledge the efforts the authors made to provide more evidence, and agree with the authors that these experiments are challenging. However, without solid experimental evidence, the authors should, at least, tone down the claim they are making. Moreover, for Fig. R2, it is interesting that this beta-hairpin folds on its own. More biophysical characterization will be required to confirm the folding of this hairpin.

Response to referee comments

Referee 1:

In the revised manuscript, Mojumdar et al carried out additional experiments and attempted to address the questions I raised in the first round of review. However, I remain unconvinced by the response and additional experimental evidence provided by the authors.

1) With regard to the ΔL_c analysis: although the cumulative ΔL_c analysis could be potentially advantageous in certain cases, a fundamental assumption of this method is that protein unfolding follows a particular sequence of events. This assumption is unfounded. An example in this regard can be found in Stigler et al Science, 2011, 334, 512. For a protein that is prone to misfolding, it is hard to imagine that misfolding will follow a particular well-defined pathway. Without proving the validity of this assumption, my concern remains: “Reporting cumulative ΔL_c may lead to wrong grouping of the events. I suspect that the wide ΔL_c distribution for some intermediate states is simply due to the grouping of different and unrelated intermediate states into one population.”

There are three main issues raised in this comment to discuss: (i) the notion that analyzing the cumulative ΔL_c requires one to assume that the protein unfolding follows a particular sequence of events, (ii) the idea that using cumulative ΔL_c may lead to “wrong grouping of events” and that the peaks in the distribution may include contributions from more than one state (this is the core concern), and (iii) the implicit claim that it would be better to analyze the incremental ΔL_c instead. We address each of these issues in turn:

(i) The referee asserts that in order to analyze the cumulative ΔL_c one must make a “fundamental assumption” that the folding pathway follows a specific sequence of events. We respectfully note that it is this assertion that is unfounded, not our analysis—there is no requirement to make any such assumption to perform an analysis of the cumulative ΔL_c , nor do we do so in our work. Indeed, we agree with the referee that such an assumption is not generally warranted: whereas some proteins appear to follow specific sequences of events (*e.g.*, see work by Englander discussing this issue, Englander & Mayne *PNAS* (2017) 114:8253-8258), others appear not to do so (as in the example cited by the referee).

Our analysis of the cumulative ΔL_c involves two pieces: first we identify the minimum number of states that must be present to account for the peaks that we see in the distribution, and second we determine the most likely sequences of partially-folded structures that match the observed data. We specifically do *not* impose any requirement that a particular sequence of events must transpire in the folding (*e.g.*, sequential unfolding from one end to another), nor indeed that there even be a single pathway; instead, we test out all possible combinations of intermediates where parts of the native structure have unfolded in order to search for all possible pathways through these states that are consistent with the pulling curves. As stated in the manuscript (bottom of page 7): “we considered all possible combinations of folded and unfolded strands consistent with the topology of the native fold (Table S1), including not only cases where strands were peeled from the edges of the structure but also those where the native fold ruptured internally (leading to states with disjunct structural elements).” Fig S1 includes a concrete example of such a structure (with strands 2&3 folded in one part, and strands 5 through 8 folded separately); this structure does not connect to any of the other structures that could explain the other intermediate states in that pulling curve, which is why it isn’t included in the model of the folding (Fig 3B).

The fact that we find the relatively simple model presented in Fig 3B explains the vast majority of the data is thus an actual result, not something that was pre-ordained because of assumptions built into the analysis. We fully acknowledge that the picture may be more complicated than what we present in Fig 3B—indeed, we stated (page 8) that the FECs “inconsistent with this model might reflect additional pathways.” In addition, some FECs are consistent with more than one pathway. Our analysis aims only to find the least-complex model that can account for the data, as is the standard approach in science (“Ockham’s razor”). We have revised the Methods section (page 21) to emphasize that our “analysis was used to identify the minimal set of pathways able to explain the data.”

(ii) The core concern of the referee regarding the use of the cumulative ΔL_c is that we may be mis-identifying the intermediate states because different structures may end up ‘grouped’ into a single peak in the distribution. We agree that this is an important issue, but we respectfully suggest that the referee’s criticism is misplaced, as we have already explicitly addressed this concern in the previous revision of the manuscript through our folding pathway ‘network’ analysis (Fig S1). The critical point is that, even though we identify a minimum of 8 distinct intermediate states from the distribution of cumulative ΔL_c , we allow for the fact that more than 1 structure might contribute to any given peak in the distribution, exactly as the referee posits.

We illustrated this analysis in Fig S1 for one particular pulling curve containing 3 intermediates. As seen in this figure, more than one different structure is consistent with the observed contour length for each of the intermediate states; in the case of the 2nd intermediate, we identified 5 possible structures that could explain the observed cumulative ΔL_c at that point in the pulling curve. Nevertheless, the network analysis identified only one pathway through all of these possibilities (subject to the physical constraints that we described in the manuscript) in the case of this pulling curve. For other pulling curves, more than one pathway may have been consistent with the data; see for example Fig RR1 below. Overall, the model we proposed was able to account for the great majority (>85%) of pulling curves, making it the simplest explanation of

Figure RR1: Pathway analysis of a FEC showing 3 different pathways consistent with the observed length changes (see Fig S1 for explanation). Two of the pathways are included in the minimal model, one is not. Note that this analysis explicitly allows for the fact that different structures may produce similar lengths, and includes structures that involve unfolding in the “middle” of the protein, and is not restricted to sequential peeling from the edges.

the data. That said, however, we did find other potential pathways consistent with a small minority of the curves.

We have revised the manuscript to explain this question more clearly. We now state explicitly in the Methods section (p. 20) that the pathway analysis takes into account the possibility that a single peak in the distribution might reflect more than one structure. We also added two new figures to the supporting material: one (Fig S1B) showing an example of a FEC consistent with 3 pathways (one being a pathway not included in the consensus model of Fig 3B), and a second (Fig S2) showing some of the neglected “minority” pathways found in the network analysis and how they fit in with the consensus picture of Fig 3B.

Figure RR2: Incremental contour length change in SOD1 unfolding. Only 3 peaks are discerned, but up to 6 transitions are observed directly in FECs. The incremental ΔL_c thus obscures distinct transitions that we know must be present.

(iii) The implicit claim of the referee is that it would be better to analyze the incremental ΔL_c . As we have already argued in the previous response, the incremental ΔL_c hides key information about the order of events in the unfolding and hence reduces the amount of information that can be retrieved. To illustrate this notion concretely, we have plotted the distribution of all incremental ΔL_c values for all transitions in unfolding FECs (Fig RR2). We see that this distribution does not properly resolve all of the intermediates that are present: at most 3 peaks can be resolved, yet direct inspection of the FECs shows that some curves have as many as 6 structural transitions! The incremental ΔL_c thus obscures what is going on and provides less information than the cumulative ΔL_c .

Finally, the referee mentions that it is hard to imagine that a protein will follow a particular well-defined pathway for misfolding. We respectfully note that this comment is not

relevant, as we are not suggesting that such is the case for SOD1—to the contrary, we identify several different possible routes for misfolding in Fig 3B. We have revised the manuscript in the Discussion section (bottom p. 13) to state this point explicitly.

2) *“The data were measured from 6 different molecules, with an average of about 400 pulls each. This is enough data from each molecule to check that the histograms from different molecules are indeed similar.”*

I am concerned about the extremely low number of molecules reported in this work. Although each molecule contains ~400 pulls, this fact does not prove that these six molecules reflect the true behaviors of SOD1. More evidence/data is needed from different molecules.

We respectfully note that the referee’s assertion that our data are insufficient because we report measurements on an “extremely low number” of molecules (6) is contradicted by comparison to the literature. The top researchers in the field using optical tweezers to study protein folding often publish papers based on data from ~5–15 molecules for a given protein and solution condition—in other words, ~5–15 molecules is widely considered to be sufficient to characterize the “true behavior” of a protein in a given condition. Many publications unfortunately do not list the number of molecules explicitly, but here are a few concrete examples from leading groups where the number of molecules is mentioned:

Labs of Carlos Bustamante and Susan Marqusee, UC Berkeley:

- Elms *et al.* *PNAS* (2012) 109:3796-3801 reports measurements on 5 molecules of one variant of apomyoglobin, and 8 molecules of another variant.
- Jagannathan *et al.* *PNAS* (2012) 109:17820- 17825 reports force-jump measurements on 7 molecules, and ~180 force-extension curves on likely a similar number of molecules (number not reported explicitly), for two variants of SH3.
- Guinn *et al.* *Nature Communications* (2015) 6:6861 reports measurements from “at least” 6 different molecules of SH3 (generating “at least” 70 transitions) for each condition in the study.

Lab of Matthias Rief, TU Munich:

- Gebhardt *et al.* (*PNAS* (2010) 107:2013-2018) report measurements on 11 molecules of a leucine zipper.
- Zoldak *et al.* (*PNAS* (2013) 110: 18156-18161) report ~450 pulling curves measured on “several” (presumably less than ~12, else other descriptions like “a dozen” would have been used) molecules for each of two constructs.
- Austen *et al.* (*Nat Cell Biol* (2015) 17:1597-1606) report calibrations of two protein-based force sensors using ~340 pulls on 15 different molecules each.

Lab of Yongli Zhang, Yale University:

- Xi *et al.* (*PNAS* (2012) 109:5711-5716) report measurements on 14 molecules of the leucine zipper pIL. Note that this paper investigates misfolding in the protein, too.

Labs of Matthew Lang and Susan Lindquist, MIT:

- Dong *et al.* (*Nat Struct Mol Biol* (2010) 17:1422-1430) report pulling measurements on ~10–30 prion fibril tethers per condition. Note that each fibril could only be measured once, hence the minimum number of molecules needed to obtain sufficient statistics was a bit higher (which made this experiment more challenging than normal).

Lab of Sander Tans, AMOLF:

- Jakobi *et al.* (*Nature Communications* (2011) 2:385) report measurements on 6 molecules of von Willebrand factor for each of two conditions.
- Mashagi *et al.* (*Nature* (2013) 500:98-101) report 55 pulls of a MBP tetramer with the chaperone trigger factor present, measured from ~5–10 molecules (since it is reported that up to 10 pulls were possible from a single molecule).

As this brief survey shows, the number of molecules we have measured is within the range that is often reported and considered sufficient to characterize the “true behavior” of a protein. Nevertheless, it is near the low end of the range (in terms of number of molecules, not the number of pulling curves), and we are sensitive to the referee’s wish to include additional molecules for a more robust characterization. We have therefore added data from a few more molecules to the analysis of the intermediates and folding pathways: an extra 838 FECs from 6 more molecules, bringing the total up to 3050 FECs from 12 molecules. Including these additional FECs in the analysis did not change the results—any differences were within the experimental uncertainty previously reported and are difficult if not impossible to discern in the figures, giving confidence that the molecules we had measured previously were not somehow unrepresentative. The agreement with the minimal model of the folding in Fig 3B was very slightly improved (accounting now for 86% of all traces, up from 82%).

We note that the referee’s view that the number of molecules in our study is “extremely low” may be influenced by a comparison to the literature where proteins are unfolded by AFM, rather than optical tweezers. However, such a comparison is not justified, because of the differences that exist between the methods: typically, with AFM one only gets a single pull per molecule, whereas with optical tweezers one gets many pulls per molecule. Hence AFM studies typically report many more molecules but fewer pulls than optical tweezers papers.

3) *Regarding experiments attempted to provide more experimental evidence for the peeling model:*

I acknowledge the efforts the authors made to provide more evidence, and agree with the authors that these experiments are challenging. However, without solid experimental evidence, the authors should, at least, tone down the claim they are making.

We assume that the claim the referee mentions here is the model we propose for the folding, summarized in Fig 3B. As noted in our response to point 1 (i) above, we had already modified the results section of the manuscript to make it clear that, although our model is the most likely explanation (consistent with the vast majority of pulling curves observed, as well as with the one truncation mutant we could measure and also the simulations), other pathways are possible. We have now further revised the manuscript to clarify this point and “tone down” the claims as requested:

- The statement in the abstract that the intermediate states correspond to the sequential formation of each β -strand in the native structure was changed to say that the intermediates are “consistent with the formation of individual β -strands in the native structure.” This formulation implies the possibility of multiple pathways, and does not assert that there is a specific sequence of events, as requested by the referee.
- We revised the Methods (pages 7, 8) and Discussion sections (bottom p. 10) to clarify that our model presents the “most likely” (rather than “dominant”) sequence of steps during SOD1 folding (indicating that other pathways are also possible, but they are not able to account for as much of the data). We also changed the claim (bottom p. 12) that the intermediates “correspond to” the sequential addition/removal of each β -strand to the statement that the length changes were “consistent with the sequential addition/removal of individual β -strands.”
- A statement explicitly acknowledging that other pathways were possible, but consistent with much less of the data, was added to the “Folding pathway analysis” section of the Methods (bottom p. 20). As a concrete illustration of this point, Fig. S1 was modified to include a second example of the network analysis (Fig S1B), in which 3 pathways were consistent with the data (two being in the minimal model of Fig 3B, the last one being a “minority” pathway that is not included in the minimal model). We also added another new figure (Fig S2) illustrating some of the low-probability but possible pathways found in our analysis that were not included in the minimal model.

We believe that the claims, as we have stated them in the revised manuscript, are fully supported by the data and analysis.

Moreover, for Fig. R2, it is interesting that this beta-hairpin folds on its own. More biophysical characterization will be required to confirm the folding of this hairpin.

We agree that a more in-depth study of this hairpin will likely prove to be interesting. However, such a study lies outside the scope of the current paper: the point of measuring this hairpin is simply to show that it is indeed independently stable, as our model of the folding implies. Had the hairpin not folded on its own, that would have disproven our model. The fact that this hairpin does fold on its own is clearly demonstrated by our measurements (Fig 4), without any need for further confirmation. Since this is the key property of the hairpin that is relevant to our argument, we believe that it is unreasonable to insist that we go beyond what is strictly necessary and provide a complete biophysical characterization of the hairpin.

List of most significant changes to the manuscript:

1. New data were added. Figures 1, 2, 3, 6, and S3 were updated to reflect the additional data, as were Tables 1 and 2. The main text was also updated to reflect the new number of molecules and pulls.
2. The abstract was modified to “tone down” the conclusions about the intermediate states.
3. Results, p. 8: the analysis of intermediate states was described more clearly.
4. Results, p. 8 and Discussion, p. 10: we qualified our model of the intermediates as “the most likely” explanation of the data.
5. Discussion, p. 12: the conclusions about the intermediate structures were “toned down”.
6. Discussion, p. 13: we added an explicit statement that there are multiple possible misfolding pathways.
7. Methods, p. 20: the fact that the analysis allows for more than one structure contributing to a given peak length is now stated explicitly, to clarify the analysis.
8. Methods, p.20-21: a statement noting that the model is the minimal explanation and other pathways exist was added, as were two additional figures: Fig. S1B (showing an example of the analysis where multiple pathways were consistent with a single FEC) and Fig 2 (showing 2 examples of “minority” pathways not included in the minimal model).